# Raman microscopy-based quantification of the physical properties of intracellular lipids

Masaaki Uematsu [1,2✉] & Takao Shimizu[1,3]

The physical properties of lipids, such as viscosity, are homeostatically maintained in cells and are intimately involved in physiological roles. Measurement of the physical properties of plasma membranes has been achieved primarily through chemical or genetically encoded fluorescent probes. However, since most probes target plasma membranes, physical properties of lipids in intracellular organelles, including lipid droplets (LDs) are yet to be analyzed. Here, we present a novel Raman microscopy-based approach for quantifying the physical properties of intracellular lipids under deuterium-labeled fatty acid treatment conditions. Focusing on the fact that Raman spectra of carbon-deuterium vibration are altered depending on the surrounding lipid species, we quantitatively represented the physical properties of lipids as the gauche/trans conformational ratio of the introduced labeled fatty acids, which can be used as an indicator of viscosity. Intracellular Raman imaging revealed that the gauche/trans ratio of cytosolic regions was robustly preserved against perturbations attempting to alter the lipid composition. This was likely due to LDs functioning as a buffer against excess gauche/trans ratio, beyond its traditional role as an energy storage organelle. Our novel approach enables the observation of the physical properties of organelle lipids, which is difficult to perform with conventional probes, and is useful for quantitative assessment of the subcellular lipid environment.

[1] Department of Lipid Signaling, National Center for Global Health and Medicine, 1-21-1 Toyama, Shinjuku-ku Tokyo, Japan. [2] Department of Lipidomics, Graduate School of Medicine, The University of Tokyo, 7-3-1 Hongo, Bunkyo-ku Tokyo, Japan. [3] Institute of Microbial Chemistry, 3-14-23 Kamiosaki, Shinagawa-ku Tokyo, Japan. ✉email: uematsu@m.u-tokyo.ac.jp

Forming the boundaries of cells and organelles, lipids are a fundamental component of all living organisms. In eukaryotic cells, their physical properties are homeostatically maintained inside cells mainly via the tuning of phospholipid acyl chain composition[1,2]. In general, a higher degree of acyl chain unsaturation increases membrane flexibility. The physical properties of lipids are also involved in various physiological roles, such as enzymatic activities and signal induction, through the regulation of membrane protein diffusion or membrane rigidity[3–5]. As such, various probes and methods have been developed to quantify the physical properties of biological membranes at the subcellular level, predominantly through fluorescence techniques[6,7]. However, these methods require the introduction of chemically synthesized or genetically encoded fluorescent probes, thereby cannot avoid the artificial effect of introduced probes themselves on the physical properties of the membranes being analyzed. In addition, the majority of probes mainly target plasma membranes, and the other intracellular structures are yet to be analyzed. In particular, physical properties of lipid droplets (LDs), which are mainly composed of triglyceride, were hardly quantified nor studied due to the lack of appropriate probes, even though nearly all eukaryotic cells can create LDs and that their lipid compositions vary in response to external stimuli. Thus, it remains unclear how LDs contribute to the physical properties of the intracellular lipid environment, and they are thought to only store excess fatty acids in the form of triglycerides as an energy source[8].

Raman microscopy is a promising approach to circumvent these problems[9]. As Raman spectra reflect the vibrational modes of molecules, they provide information regarding a molecule's chemical structure, but they also have the potential to reveal other characteristics that may affect a molecule's vibrational mode. Previous in vitro studies have demonstrated that Raman spectra of carbon-hydrogen (C–H) stretch in fatty acids or phospholipids shows different shapes depending on viscosity changes caused by altered temperature or pressure[10,11]. However, it has been unknown whether the C–H stretch spectrum is sensitive enough to differentiate between the physical properties of lipids under physiological in vivo conditions. In addition, as a majority of biomolecules contain C–H bonds in their substructures, it is difficult to obtain a pure C–H stretch spectrum for intracellular lipids using Raman microscopy. Furthermore, analytical methods must be developed to convert the observed spectrum into a quantitative value that reflects the physical properties of lipids. A recent report using stimulated Raman scattering (SRS) microscopy demonstrated that C16:0(d31) (fatty acids with sixteen carbons and zero double bonds labeled with thirty-one deuterium atoms) treatment results in the formation of phase-separated endoplasmic reticulum (ER)-associated membrane domains, quantifying the physical properties of lipids inside these domains as the lateral diffusion coefficient[12]. However, their calculation of the diffusion coefficient was based on spatial-temporal analysis resulting from C16:0(d31) pulse labeling, thus whether the physical properties of lipids can be quantitatively obtained from spectral transitions is yet to be established.

Here, we present a Raman microscopy-based approach for quantifying the physical properties of subcellular lipids under deuterium-labeled fatty acid treatment using the gauche/trans conformational ratio, which can be used as an indicator of viscosity[13]. Specifically, observed Raman spectra were quantitatively converted into the gauche/trans ratio of the introduced deuterated fatty acids based on experimentally measured reference spectra in vitro and model membrane simulations in silico. This novel approach, that is, using the target lipid itself as a probe, overcame the innate problems of conventional probes affecting the physical properties of the lipid environment and

enabled the investigation of the physical properties of LDs at the same time. Applying this method to intracellular imaging revealed that while the gauche/trans ratio in LDs dynamically changed depending on perturbations attempting to alter the lipid composition, it was relatively constant in non-LD regions. This result suggests that LDs may function to buffer the intracellular gauche/trans ratio and maintain the physical properties of the lipid environment in cytoplasmic regions, beyond its traditional role as an energy storage organelle. Our quantitative evaluation will contribute to the understanding of biological functions and regulatory mechanisms associated with the physical state of intracellular lipid environments.

## Results

**Spectra of saturated fatty acids reflect the physical properties of lipids.** Raman spectra derived from C–D stretch appear in the silent region (around 1800–2600 cm$^{-1}$), where the spectra from cellular biomolecules are hardly detectable. That means the spectra of deuterium-labeled fatty acids can be observed with minimal interference from other molecules, even if they are taken up into cells. Therefore, we focused on deuterium-labeled fatty acids, one of the simplest lipids, as candidate probes for measuring the intracellular physical properties of lipids. First, we investigated whether drastic changes in the physical state of lipids, namely, changes in the liquid-solid phase of lipids, are detectable using the Raman spectra of C–D stretch from deuterium-labeled fatty acids. To this end, the standard spectra of deuterium-labeled saturated and unsaturated fatty acids (in vitro spectra) were compared with the spectra of each fatty acid from LDs after their incorporation into HeLa cells (in vivo spectra) (Fig. 1a, b). Saturated (Fig. 1a) and unsaturated (Fig. 1b) fatty acids used in this experiment are in solid (melting temperature > 60 °C) and liquid (melting temperature < 14 °C) states, respectively, at room temperature, while fatty acids incorporated into cells should be closer to the liquid state, as they are mixed with other endogenous lipids. Therefore, differences between the in vitro and in vivo spectra of labeled saturated fatty acids are expected to be larger than that of unsaturated fatty acids. Spectral comparisons were qualitatively performed by fitting reconstructed in vitro spectra (solid red lines in Fig. 1a, b) to in vivo spectra (solid black lines in Fig. 1a, b); the reconstruction was performed using the weighted sum of in vitro spectra, background spectra, and baseline drift (dashed red lines, dashed gray lines, and dashed gray straight lines, respectively, in Fig. 1a, b), as previously described[14]. As a result, the spectra of labeled saturated fatty acids (C16:0[d2], C16:0[d31], C18:0[d3], C18:0[d5], and C20:0[d4]) clearly showed poorer fitting than that of labeled unsaturated fatty acids (C18:1[d17], C18:2[d4], C18:3[d14], C18:2[d11], C20:4[d8], C20:4[d11], C20:5[d5], and C22:6[d5]).

We also confirmed the correlation between spectral transitions and the solid-liquid state of lipids in another way, using bromine (Br)-labeled saturated fatty acids C16:0(Br) and C18:0(Br), both of which exist in the liquid state at room temperature (Fig. S1a). In contrast to the results of C16:0(d2), C16:0(d31), C18:0(d3), and C18:0(d5), in vivo spectra from the carbon-bromine (C–Br) stretch of C16:0(Br) and C18:0(Br), which appears around 533 and 612 cm$^{-1}$, were very similar to in vitro spectra (Fig. S1b). These results confirm that in vivo and in vitro spectral differences represent variances in the physical state of fatty acids. The results also suggest that spectral transitions do not result from the metabolism of incorporated labeled fatty acids, as some fractions of C16:0(Br) and C18:0(Br) are known to undergo metabolic processes[15].

Further in vitro experiments were performed to determine that the spectral differences represented not only the discontinuous

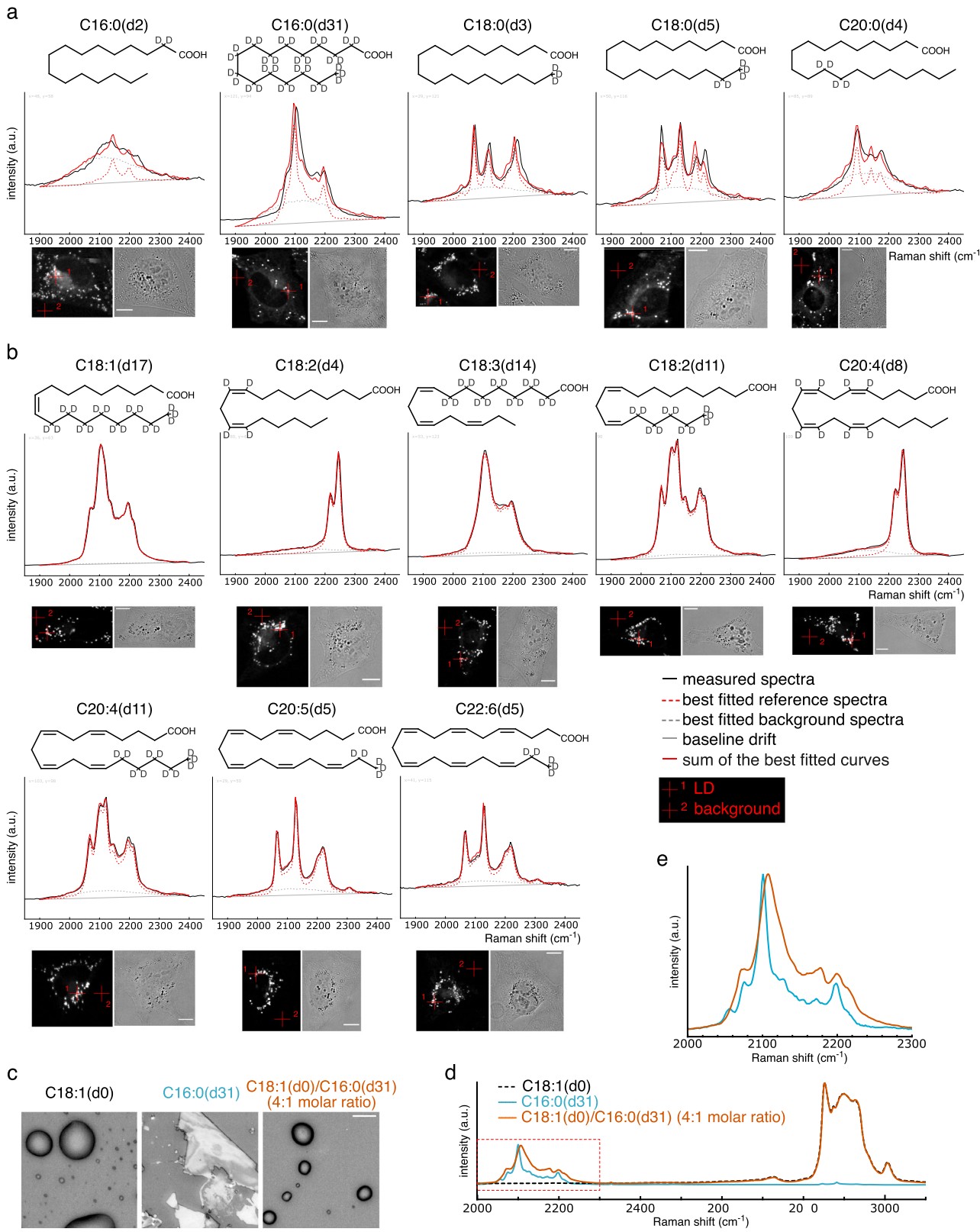

physical state of lipids, that is, solid and liquid states, but also represented continuous physical properties of lipids. For this purpose, we chose C16:0(d31), which displayed the clearest differences between in vitro and in vivo spectra, with the latter exhibiting a wider maximum peak around 2100 cm⁻¹ and a slight right shift of the peak top. While the pure C16:0(d31) in vitro was

in a solid-like state, a mixture of C18:1(d0) and C16:0(d31) in a 4:1 molar ratio appeared to exist in a liquid-like state (Fig. 1c), with spectra of the C–D stretch from C16:0(d31) also showing similar transitions to those observed in vivo (Fig. 1d, e). Notably, the spectral transitions were not discontinuous, but the gradual changes of the spectra were observed depending on the amount of

**Fig. 1 Spectra of saturated fatty acids are affected by the surrounding lipid environment. a, b** Comparison of Raman spectra of saturated (**a**) and unsaturated (**b**) fatty acid between in vivo and in vitro. HeLa cells treated with indicated deuterium-labeled fatty acids (30 μM) for 24 h were fixed and observed using Raman microscopy. Representative measured spectra at LDs (represented by red crosshair 1 in the bottom images) are shown in black lines, and the results of the best-fitted in vitro spectra, background spectra, baseline drifts, and their summations are shown in dashed red lines, dashed gray lines, dashed gray straight lines, and solid red lines, respectively. Spectra at red crosshair 2 in the bottom images were used as the background spectra for the fittings. Measurement of multiple cells and samples resulted in similar results. Scale bars indicate 10 μm. **c, d** Bright-field images and Raman spectra of C16:0(d31), C18:1(d0), and their mixture. The Y-axis of each spectrum was arbitrarily scaled to make the comparison clearer by equalizing the heights of spectra derived from C–D stretching (C16:0[d31] and the C18:1[d0]/C16:0[d31] mixture) and C–H stretch (C18:1[d0] and the C18:1[d0]/C16:0[d31] mixture). The magnified graph of the area framed by the dashed red square in (**d**) is shown in (**e**).

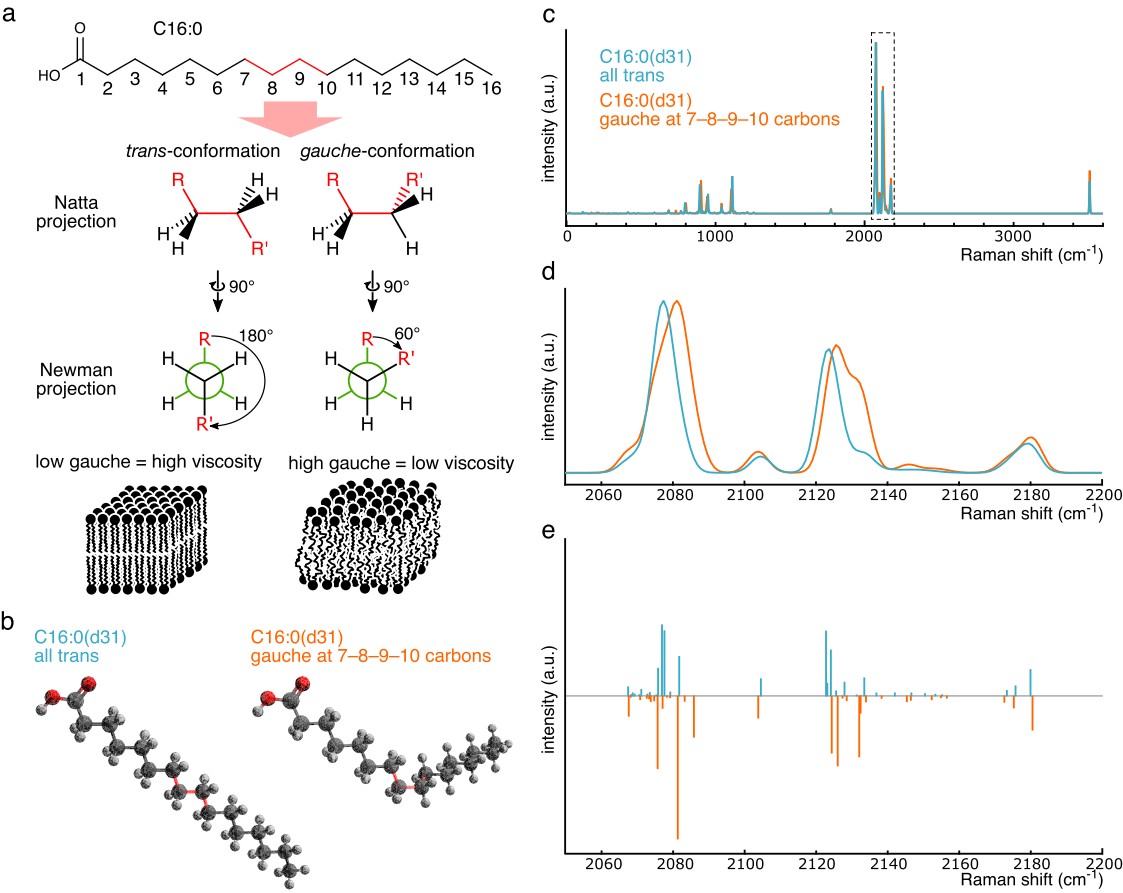

**Fig. 2 Gauche/trans conformation affects the spectra of C–D vibration. a** Schematic image of gauche/trans conformation and membrane viscosity. Gauche/trans conformation is shown with Natta and Newman projections. Lipid bilayers with a lower (left) and higher (right) gauche/trans ratio have higher and lower viscosity, respectively. The numbering of carbon atoms in C16:0 is displayed in the skeletal formula. **b** Structures of C16:0(d31) with all trans (left) and one gauche conformation at 7-8-9-10 consecutive carbons (right). Dark gray, light gray, and red atoms represent carbon, deuterium, and oxygen atoms, respectively. Chemical bonds between four consecutive carbon atoms corresponding to the skeletal formula in (**a**) are shown in red. **c** Raman spectra simulated using DFT. Magnified view of the dashed box and its corresponding Raman activity are shown in (**d**) and (**e**), respectively. The intensity was normalized by equalizing prominent peak heights of the C–D vibration region. Graphs (**d**) and (**e**) are also shown in Fig. S2a, b.

C18:1(d0) (Fig. S4a). These results indicate that the spectral transitions of C16:0(d31) reflect more than the phase differences and represent some continuous physical properties of lipids.

**Gauche/trans conformations affect the spectral shape of C16:0(d31).** Next, we investigated what kind of physical properties the spectral transitions represented. Since Raman spectra represent the vibrational modes of molecules, they can be affected by the conformational state of those molecules. In fatty acids, three interconverting conformational isomers exist for four consecutive carbon atoms (A–B–C–D): one trans and two gauche conformations (Fig. 2a). These isomers correspond to three local energy minima of the four consecutive carbon atoms:

when the dihedral angle between planes, defined by two sets of three carbon atoms (A–B–C and B–C–D), is approximately 180 and ±60 degrees. In the case of phospholipid fatty acyl groups, the ratio of the four consecutive carbon atoms with gauche conformers to those with trans conformers (referred to as the gauche/trans ratio hereafter) is closely related to the lipid packing and the viscosity. Perturbations on the physical properties by changing the temperature are known to cause changes in packing, gauche/trans ratio, and viscosity[13]. The introduction of steric hindrance by *cis*-double bonds also causes weaker lipid packing, leading to a lipid membrane with low viscosity[16,17]. At the same time, previous studies also reported the spectral transitions of Raman or infrared spectra similar to ours depending on the temperature[10,18].

Therefore, we investigated whether the spectral changes observed in Fig. 1 could be reproduced in silico by modulating the gauche/trans conformation. For this, we used density functional theory (DFT), which was developed to perform quantum mechanical modeling to calculate the electronic distribution and vibrational characteristics of compounds. Raman spectra of two types of C16:0(d31) were simulated: one in which all four consecutive carbon atoms were in trans conformation, and the other in which only one set of four consecutive carbon atoms was in gauche conformation (Fig. 2b). Simulated Raman spectra qualitatively reproduced the spectral transitions observed in Fig. 1, with increased width of the largest peak and a right shift of the peak top for 16:0(d31) with one gauche conformation (Fig. 2c–e). Similar spectral changes were confirmed for most of the identical simulations performed for C16:0(d31) with one gauche conformation at varying positions (Fig. S2). These results suggest that differences in gauche/trans conformations affect the spectral shape, confirming that C16:0(d31) is suitable as a probe to monitor the gauche/trans ratio of the lipid environment.

Notably, although in silico analyses reproduced qualitative spectral transitions, they were unable to replicate the exact shape of experimentally observed spectral transitions, as shown in Fig. 1 and Fig. S4a. One possibility for this discrepancy is that one C16:0(d31) molecule contains thirteen positions that can possess a gauche/trans conformation, and that in vitro Raman spectra are derived from a mixture of heterogeneous conformational isomers containing various combinations of gauche/trans conformations at each position. Since our in silico simulation calculated the spectra of only one conformational isomer of one molecule at a time, it was challenging to reproduce the exact spectra of such a heterogeneous population of lipid molecules.

**Spectra of C16:0(d31) are sufficiently sensitive to detect physical properties of different lipid environments in vivo.** Thus far, in vitro and in silico analyses have demonstrated that C16:0(d31) is a promising probe for detecting the physical properties of lipid environments. Next, we assessed its application in vivo to observe the intracellular lipid environment under moderate physiological changes. Hence, we attempted to capture the in vivo spectral transitions of C16:0(d31) incorporated into HeLa cells associated with the fatty acid composition and subcellular localization. HeLa cells were incubated with different compositions of fatty acids: C16:0(d31) mixed with C16:0(d0) or C20:4(d0) (30 μM each) (Fig. 3a). Then, spectra in LD and non-LD regions were compared for each condition by subtracting the background spectra drawn with gray lines (see Methods section for details). The results showed a rightward shift of the peak top of the spectrum in LDs of HeLa cells treated with the C16:0(d31)/C20:4(d0) mixture, compared to that with the C16:0(d31)/C16:0(d0) mixture, with the peak width widening (the inset in the top-right graph in Fig. 3a). These results suggest that the high degree of C20:4(d0) unsaturation resulted in a higher gauche/trans ratio of incorporated C16:0(d31) inside LDs. In contrast, such spectral transitions were hardly observed when comparing spectra of non-LD regions (the inset in the bottom-right graph in Fig. 3a). As for spectral differences between different subcellular locations, that is, LD and non-LD regions, the spectra of LDs showed a lower C16:0(d31) gauche/trans ratio than that of non-LD regions when treated with the C16:0(d31)/C16:0(d0) mixture (the inset in the bottom-left graph in Fig. 3a), while treatment with the C16:0(d31)/C20:4(d0) mixture resulted in a slightly higher gauche/trans ratio in the LD region (the inset in the bottom-middle graph in Fig. 3a).

Similar spectral transitions were also observed for HeLa cells treated with different concentrations of C16:0(d31) (Fig. 3b).

Under the condition treated with lower concentrations (10 μM) of C16:0(d31), spectra derived from LDs were wider and had a rightward-shifted peak top compared to the spectra of LDs treated with higher concentration (50 μM) of C16:0(d31), suggesting a higher gauche/trans ratio (the inset in the top-right graph in Fig. 3b). In contrast, spectra of non-LD regions were relatively constant between the two conditions (the inset in the bottom-right graph in Fig. 3b). When spectra from LD and non-LD regions were compared in the same cells, non-LD regions showed a higher gauche/trans ratio than LDs in HeLa cells treated with 50 μM of C16:0(d31) (the inset in the bottom-left graph in Fig. 3b), while such spatial differences could hardly be observed in HeLa cells treated with 10 μM of C16:0(d31) (the inset in the bottom-middle graph in Fig. 3b).

These results provide methodological significance, as well as biological insight, into the function of LDs. The conformational state of C16:0(d31) in LDs was found to be diverse depending on the amount and composition of treated fatty acids; hence, it can be speculated that LDs are comprised of endogenous unsaturated fatty acids as well as excess C16:0(d31) taken up from the medium. Meanwhile, although the spectra of LD regions were highly dynamic in both Fig. 3a, b, those of non-LD regions were relatively constant, suggesting that the lipid environment in the non-LD region is robustly preserved probably by LD serving as a buffer for excessive amounts of fatty acids inside cells.

**Generation of reference spectra to quantify spectral transitions.** Although we have qualitatively discussed the subcellular gauche/trans ratio based on the spectral transitions of C16:0(d31), a quantitative analysis is required for the more accurate evaluation of the intracellular lipid environment. Thus, we next developed a system to quantitatively convert the spectral transitions of C16:0(d31) into gauche/trans ratio values. The quantification procedure comprises the following two steps: translation of the measured in vivo spectrum into the compositional information of the phospholipid mixture based on its in vitro reference spectra, and the calculation of the gauche/trans ratio of the in vitro phospholipid mixture using in silico model membrane simulations (Fig. 4a).

For the first step, we searched for a suitable combination of lipid species that could reproduce the in vivo spectral transitions observed in Fig. 3. In addition to the free form of C16:0(d31) used for treatment in the previous section, two types of lipids containing C16:0(d31) in their substructures were prepared and investigated: lysophosphatidylcholine (LPC) 16:0(d31), and phosphatidylcholine (PC) 16:0(d62) (Fig. S3). C16:0(d31), LPC16:0(d31), and PC16:0(d62) were mixed with C18:1(d0), C18:1(d0), and PC18:1(d0), respectively, at various ratios, and their Raman spectra were observed. The results showed that the PC16:0(d62) and PC18:1(d0) mixture could mimic the spectral transitions observed in vivo well (Fig. 4b), where the PC18:1(d0)/PC16:0(d62) ratio equaling 0 and 8 showed almost the same in vitro spectra as the LD spectra in HeLa cells treated with C16:0(d31)/C16:0(d0) (blue lines in Fig. 4b) and C16:0(d31)/C20:4(d0) mixtures (orange lines in Fig. 4b) from Fig. 3a, respectively. In addition, spectra gradually changed depending on the PC18:1(d0)/PC16:0(d62) ratio, with these intermediate spectra also reproducing the shape of in vivo spectra. These high spectral similarities indicate a high degree of resemblance in the conformational state of C16:0(d31) incorporated into cells in vivo and the fatty acyl chain of PC16:0(d62) in vitro; thus, these in vitro spectra from PC mixtures were chosen as the reference for evaluating in vivo spectra.

The mixture of free fatty acids C16:0(d31) and C18:1(d0) also showed spectral transitions (Fig. S4a). However, despite various

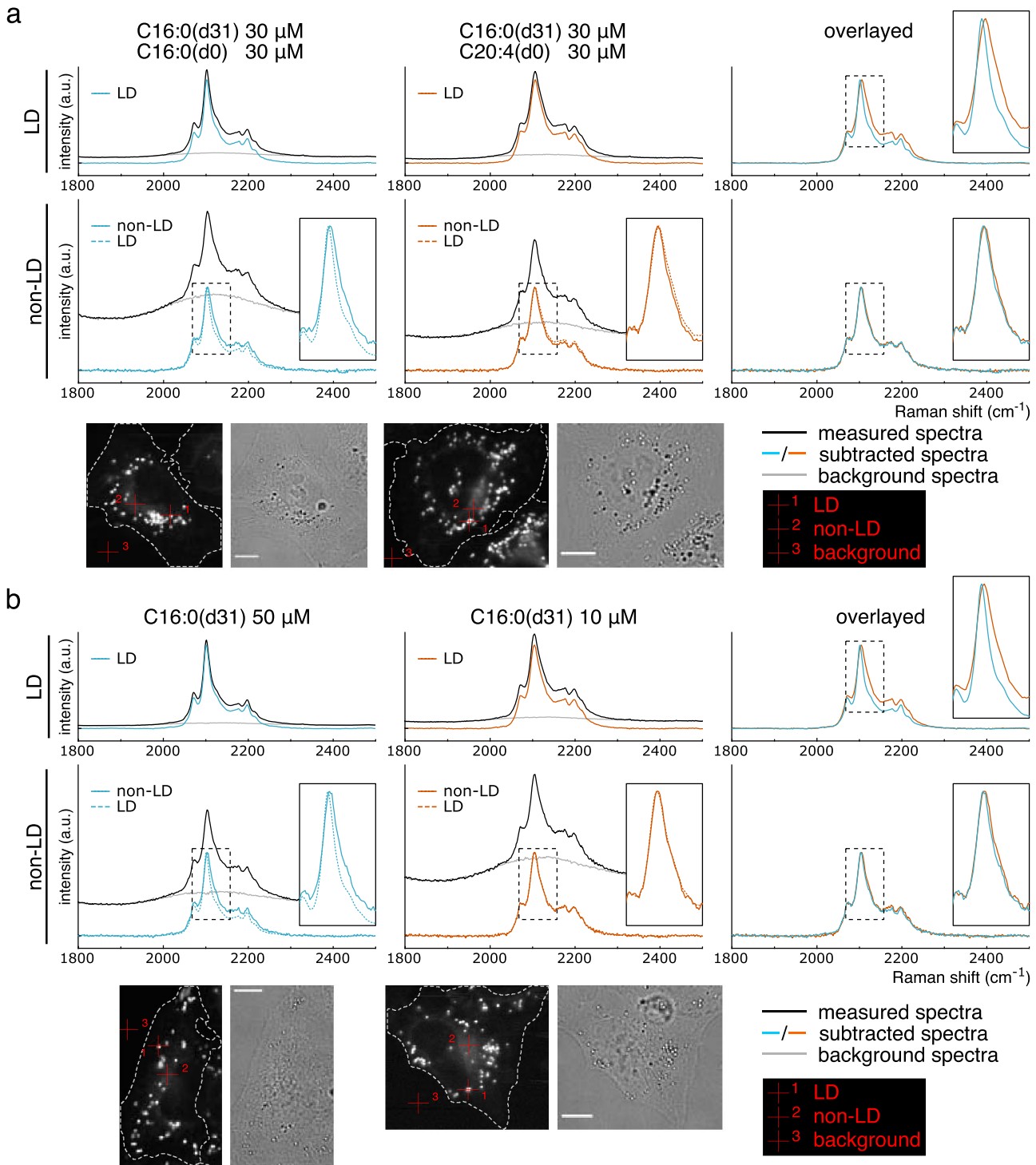

**Fig. 3 Spectral transitions of C16:0(d31) in HeLa cells. a, b** HeLa cells were treated with indicated concentrations of fatty acids or a fatty acid mixture, and the representative Raman spectra from C–D vibration were compared at LD and non-LD regions. Background spectra (gray lines) were subtracted from each measured spectrum (black lines) with the appropriate weightings (see Methods section for details) to illustrate a clear comparison, thus, producing subtracted spectra (solid blue or orange lines). Spectra from LD regions are also displayed with those from non-LD regions using dashed blue or orange lines at the bottom-left and bottom-middle graphs. Magnified views of the dashed black boxes are shown in the inset of some graphs. Pixels of representative LD regions, non-LD regions, and pixels used for background spectra are represented by red crosshairs 1, 2, and 3, respectively, in the bottom images. Two independent experiments confirmed the results. Scale bars indicate 10 μm.

adjustments of the mixing ratio of these fatty acids, the in vivo spectra of LDs in HeLa cells treated with the C16:0(d31)/ C16:0(d0) mixture (blue lines in Fig. S4a) could not be reproduced. In addition, the reproducibility of in vitro spectra was poor when the C18:1(d0)/C16:0(d31) ratio was equal to 2,

showing various patterns depending on the sample positions. These results indicate that C18:1(d0) and C16:0(d31) were not fully mixed. In contrast, the LPC16:0(d31) and C18:1(d0) mixture showed spectral changes similar to that of the PC mixture, also reproducing the in vivo spectra well (Fig. S4b). Nevertheless,

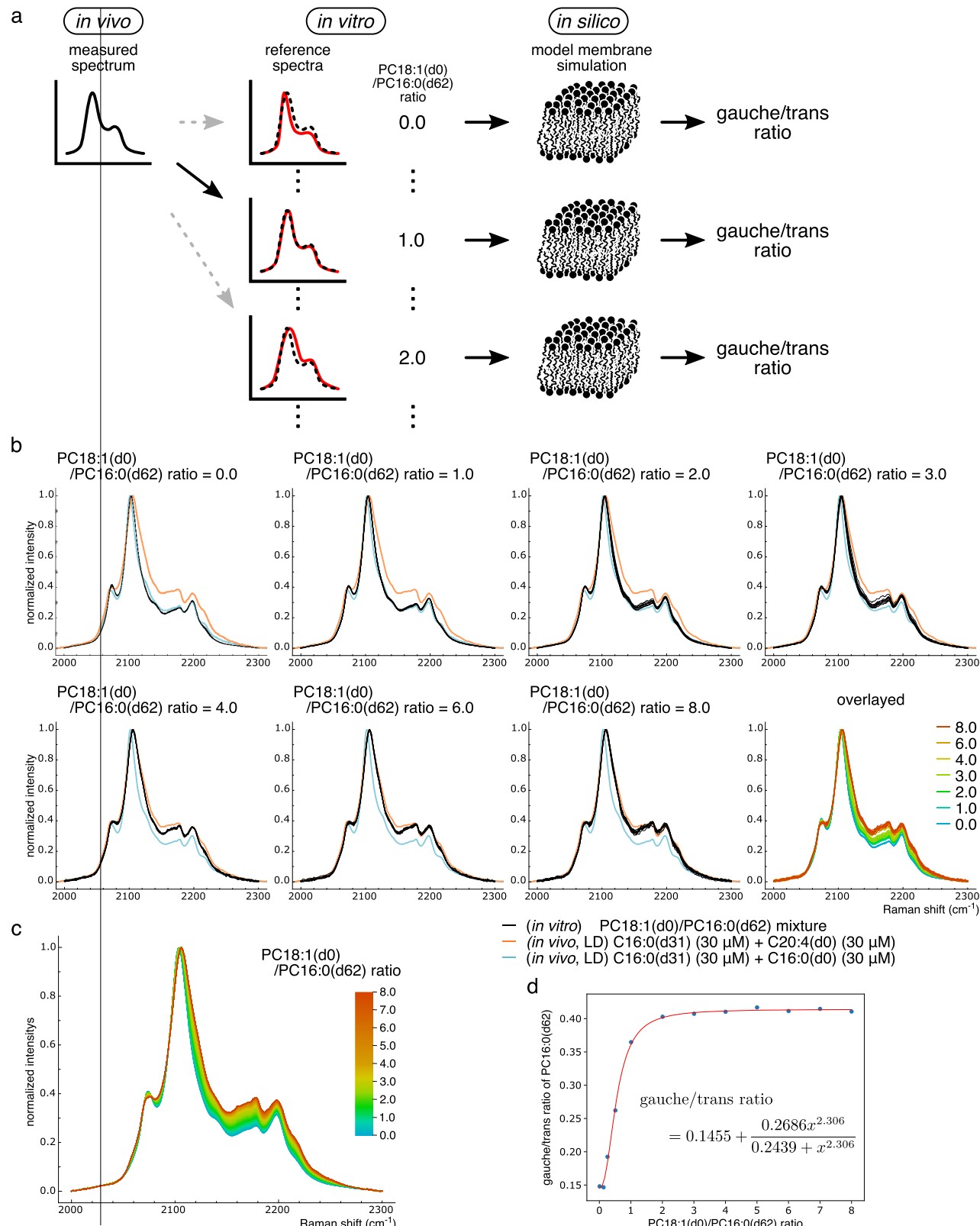

considering that the LPC16:0(d31) spectra changed relatively steeply depending on the amount of C18:1(d0) (Fig. S4c, source data are available in Supplementary Data 3), and that PCs are more abundant than LPCs in cells, spectra from PC mixtures were chosen as the reference.

Next, reference spectra were generated against continuously varying PC18:1(d0)/PC16:0(d62) ratios. Briefly, the raw spectra of

phospholipid mixtures were measured with ratio ranges from 0 to 8 at finer intervals of 0.5, then obtained spectral data sets were interpolated. First, spectra intensities were normalized, and the intensities at each spectral data point along the wavenumber axis were plotted against the PC18:1(d0)/PC16:0(d62) ratio (Fig. S5, source data are available in Supplementary Data 4 and 5). Next, piecewise cubic Hermite interpolation polynomial (PCHIP)-

**Fig. 4 Quantification of measured spectra with the PC18:1(d0)/PC16:0(d62) and gauche/trans ratios. a** Schematic image of the procedure for quantifying in vivo spectrum. Reference spectra of C–D stretch (solid red lines) were first prepared from PC18:1(d0)/PC16:0(d62) phospholipid mixtures in vitro. Next, the PC18:1(d0)/PC16:0(d62) ratio that best explains the measured cell spectrum in vivo (solid black and dashed black lines) was estimated. The obtained ratio value was further converted into a gauche/trans ratio using the MD simulation of a model phospholipid membrane consisting of the same ratio of PC18:1(d0) to PC16:0(d62) in silico. Note that only discrete ratio values are shown in the scheme, but they were actually interpolated to yield continuous values. **b** In vitro spectra of PC18:1(d0)/PC16:0(d62) phospholipid mixture used to generate the reference spectra. Phospholipids were mixed in the indicated ratio, and the Raman spectra were measured. For each condition, spectra were measured from 9 or 10 different locations of the sample and are displayed in the same graph using black lines (seven graphs excluding the bottom right one). To show the similarity between in vitro and in vivo spectra, in vivo spectra of LDs from Fig. 3a are also displayed with blue and orange lines. The bottom right graph is an overlaid graph of the other seven graphs. **c** Reference spectra generated by the interpolation of in vitro spectra displayed in (**b**). Note that, color bar scale is not the same as in (**b**). **d** Relationship between the gauche/trans ratio and the PC18:1(d0)/PC16:0(d62) ratio estimated using in silico simulation. Results of the MD simulations are displayed as blue dots, and the red line indicates the function to express the gauche/trans ratio by the PC18:1(d0)/PC16:0(d62) ratio x. The function was obtained by the optimization of parameters based on the kinetic model (see Methods section for details). Source data are available in Supplementary Data 1.

based regression was performed to obtain a continuous function from discrete data points, thereby generating spectra of the PC18:1(d0)/PC16:0(d62) mixture at an arbitrary ratio (Fig. 4c).

**Molecular dynamics (MD) simulation for the calculation of quantitative gauche/trans ratio values.** For the second step of the quantification of the spectral transition, the gauche/trans ratio of acyl chains in PC16:0(d62) was calculated using MD simulations. Phospholipid model bilayer membranes with a PC18:1(d0)/PC16:0(d62) ratio of 0.0, 0.125, 0.25, 0.5, 1.0, 2.0, 3.0, 4.0, 5.0, 6.0, 7.0, and 8.0 were prepared in silico, and the MD simulations were performed. As expected, the gauche/trans ratio increased as the relative concentrations of PC18:1(d0) increased, eventually reaching just over 0.4 (blue dots in Fig. 4d). Simulated data were fitted to an equation that was calculated based on a kinetic model (see Methods section for details), thereby allowing us to convert the PC18:1(d0)/PC16:0(d62) ratio into a gauche/trans ratio (red line in Fig. 4d).

Using this two-step quantification approach, observed in vivo spectra can be converted into a PC18:1(d0)/PC16:0(d62) ratio, and subsequently into a gauche/trans ratio.

**Quantification of spectral changes in vivo.** We applied the developed quantification method to the in vivo data. Briefly, spectral data at each pixel were converted into the PC18:1(d0)/PC16:0(d62) ratio values that account for the in vivo spectrum with the minimum residual sum of squares (RSS), considering background spectra and baseline drifts (see Methods section for details). Then, PC18:1(d0)/PC16:0(d62) ratio images were converted into gauche/trans ratio images using the function to convert the PC18:1(d0)/PC16:0(d62) ratio into a gauche/trans ratio obtained from the results of the MD simulations and the kinetic model.

First, we employed this quantification method to analyze the results of a disruption assay, confirming that the altered shape of the Raman spectrum in Fig. 3 is not a result of the metabolism of incorporated C16:0(d31), but the result of changes in the physical properties of lipids. Specifically, we treated HeLa cells with weak detergent (digitonin) to loosen the lipid packing and increase the gauche/trans ratio, without destroying subcellular lipid structures. The results of the quantified Raman imaging data showed the increased PC18:1(d0)/PC16:0(d62) and gauche/trans ratios in non-LD regions by the digitonin treatment (Fig. S6, source data are available in Supplementary Data 6). These results suggest that the observed spectral transitions reflect changes in the physical properties of the lipids, in particular increases in the gauche/trans ratio caused by the weakened lipid packing. In contrast, PC18:1(d0)/PC16:0(d62) and gauche/trans ratios did not change

in LD regions, suggesting that the LD regions are more resistant to the detergent treatment than the non-LD regions. Considering that LDs are reported to provide a protective environment that minimizes oxidation of polyunsaturated fatty acid (PUFA) to suppress reactive oxygen species (ROS) levels[19], they may have resistant characteristics against the external environment.

Next, we analyzed the in vivo data shown in Fig. 3 by using the developed quantification method. The generated PC18:1(d0)/PC16:0(d62) ratio images showed relatively uniform intensity in LD and non-LD regions, thus, confirming that the pixels analyzed in Fig. 3 reflect the region-wide physical properties of lipids (Fig. 5a, b). Representative fitting results at the same pixels as in Fig. 3 were also shown. The sum of the estimated spectra (dashed blue and orange lines) fitted well with the measured spectra (solid black lines), suggesting that in vitro spectra are a good approximation of in vivo spectra. For the gauche/trans ratio images converted from PC18:1(d0)/PC16:0(d62) ratio images, the results also showed uniform gauche/trans ratios in LD and non-LD regions (Fig. 5c, d).

The trends of the physical properties in LD and non-LD regions were further analyzed by calculating the average PC18:1(d0)/PC16:0(d62) and gauche/trans ratio of each region over multiple cells. In addition to the four conditions described in Fig. 5a–d, the following conditions were introduced to the analysis: treatment with the fatty acid mixture with a higher proportion of C20:4(d0), and pharmacological treatment using an inhibitor of stearoyl-CoA desaturase 1 (SCD1) (Fig. 5f, g, except for the rightmost condition in the non-LD region). Note that, SCD1 is an enzyme that introduces a double bond to the carbon at position 9 of C18:0-CoA and C16:0-CoA to produce C18:1-CoA and C16:1-CoA, respectively[20]. For the first condition, HeLa cells were treated with a C16:0(d31)/C20:4(d0) mixture (10 μM/40 μM); for the second condition, cells were treated with a mixture of 10 or 50 μM of C16:0(d31) and SCD1 inhibitor. As expected, PC18:1(d0)/PC16:0(d62) and gauche/trans ratios were the highest among all conditions in HeLa cells treated with C16:0(d31)/C20:4(d0) (10 μM/40 μM) for both LD and non-LD regions. As for SCD1 inhibitor treatment, PC18:1(d0)/PC16:0(d62) ratios slightly decreased in LD and non-LD regions of HeLa cells supplemented with 10 μM of C16:0(d31) and in non-LD regions of HeLa cells supplemented with 50 μM of C16:0(d31), compared to in the absence of SCD1 inhibitor. These results quantitatively detected the effect of SCD1 inhibition on the subcellular lipid environment, indicating that non-LD regions of HeLa cells treated with 10 μM of C16:0(d31) containing an SCD1 inhibitor were affected to the same extent as those treated with 50 μM of C16:0(d31) in the absence of an SCD1 inhibitor.

Notably, the PC18:1(d0)/PC16:0(d62) and gauche/trans ratios of LD regions changed dramatically in these seven conditions,

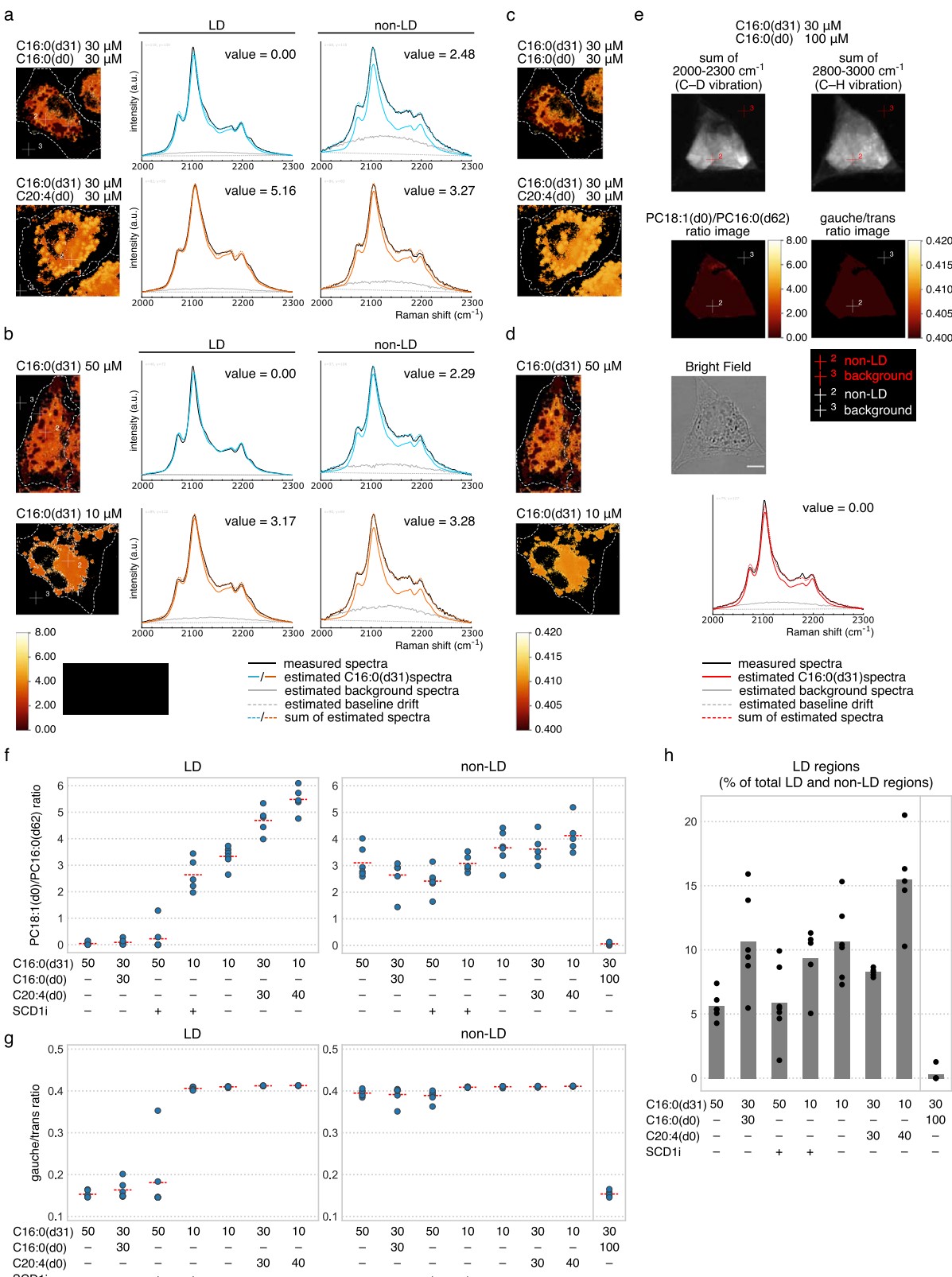

with average ratio values ranging from 0.045–5.48 and 0.153–0.413, respectively, while lipid environments in non-LD regions were relatively conserved, with ratio values ranging from 2.42–4.12 and 0.390–0.411, respectively. These results suggest the possible mechanism that keeps the physical properties of the cytoplasmic lipid environment stable, probably by LDs acting as buffers for excessive degrees of the gauche/trans ratio. To explore

the contribution of LDs to this robust lipid environment in non-LD regions, we searched the conditions under which this robustness is lost. As a result, we found that the treatment with a higher amount of C16:0 (130 μM in total) greatly reduced the PC18:1(d0)/PC16:0(d62) and gauche/trans ratios in non-LD regions (Fig. 5e–g). Meanwhile, almost no LD formation was observed under this condition (Fig. 5h), and cell viability was

**Fig. 5 Quantification of in vivo spectra revealed the preserved physical properties in non-LD regions. a, b** Raman hyperspectral data in Fig. 3 were converted into PC18:1(d0)/PC16:0(d62) ratio images. Representative results of the fitting at the same pixel as Fig. 3 (represented by white crosshairs 1 and 2) are displayed in the graphs on the right, with the calculated PC18:1(d0)/PC16:0(d62) ratio values. The background spectra from the same pixel as Fig. 3 (represented by white crosshair 3) were used for calculation. **c, d** PC18:1(d0)/PC16:0(d62) ratio images in (**a**) and (**b**) were further converted into gauche/trans ratio images. **e** HeLa cells treated with a total of 130 μM C16:0. The representative result of the fitting at the pixel represented by red and white crosshair 2 is displayed in the bottom graph, with the calculated PC18:1(d0)/PC16:0(d62) ratio value. The background spectra from the pixel represented by red and white crosshair 3 were used for calculation. The PC18:1(d0)/PC16:0(d62) and gauche/trans ratio images are displayed with the same color scaling as (**a**), (**b**), (**c**), and (**d**). The C–D and C–H vibration images and the bright field image are displayed to show the distribution patterns of C16:0(d31) and the shape of the cell. In the graph of (**a**), (**b**), and (**e**), measured spectra are indicated with black lines, and estimated C16:0(d31) spectra, background spectra, baseline drift, and their summation are displayed with solid-colored lines, solid gray lines, dashed gray straight lines, and dashed colored lines, respectively. **f, g** Comparison of quantified PC18:1(d0)/PC16:0(d62) and gauche/trans ratios between LD and non-LD regions. The average ratio of LD and non-LD regions were calculated for five to seven cells and plotted. Red horizontal dashed lines indicate the average value for each condition. Conditions are ordered according to the average PC18:1(d0)/PC16:0(d62) ratio value in the LD region. **h** Quantification of the ratio of LD area to the sum of LD and non-LD area. These ratio values were calculated using the same data set as (**f**) and (**g**), and average values and each data point are represented with gray bar-graphs and black dots, respectively. Scale bar indicates 10 μm. Source data are available in Supplementary Data 2.

greatly reduced at the same time. These results strongly suggest that the normal function of LDs to accumulate excess fatty acids is crucial for the maintenance of the lipid environment in non-LD regions, and consequently is necessary for cell survival.

To infer the mechanism of the robustness of gauche/trans ratio in the non-LD region, we also biochemically analyzed the amount of deuterated and endogenous C16:0 under the same conditions as Fig. 5f–h, in addition to the control condition (Fig. S7, source data are available in Supplementary Data 7). Briefly, extracted lipids from cells were separated into the neutral lipid and phospholipid fractions, and the amount of C16:0(d0) and C16:0(d31) incorporated in each form of lipids were measured using gas chromatograph-mass spectrometry (GCMS). While LDs contain phospholipids only in a monolayer on its surface, non-LD regions contain many tubular and vesicular membrane structures composed of phospholipid bilayers. Therefore, the C16:0 amount in phospholipid fraction should mainly reflect that in non-LD regions. As a result, more than half of C16:0(d31) taken up into cells was distributed to the phospholipid fraction for every condition. In addition to this, the total amount of C16:0 in the phospholipid fraction was not changed significantly, while that in the neutral lipid fraction was dramatically altered for conditions with 30 μM of C16:0(d31) or higher. These results exclude the possibility that the robustness in non-LD regions is achieved by restricting the distribution of incorporated fatty acids to only LD regions to prevent their translocation to non-LD regions. Instead, cells may actively incorporate external fatty acids into both LD and non-LD regions, and the robustness in non-LD regions may be maintained through the regulation of lipid exchange between LD and non-LD regions.

## Discussion

In the present study, we developed a Raman microscopy-based method for monitoring the subcellular physical properties of a lipid environment as a gauche/trans ratio. Gauche/trans ratios and lipid viscosity are intimately related with one another, with their correlations strongly suggested through experiments manipulating viscosity at different temperatures[13,18]. Thus, although a direct relationship was not demonstrated in this study, our method may be useful to indicate the viscosity of a lipid environment.

A fundamental difference between our approach and previous techniques is that our method uses the lipid itself as a probe, thus enabling us to directly measure the physical properties of lipids without causing disruption by the introduction of probes themselves. Although our method is limited to fatty acid-treated conditions, such perturbations are commonly used to study

intracellular lipid transportation and LD functions, and should therefore allow us to study these processes under more natural conditions compared to other existing techniques. In addition, the increase of the sensitivity of Raman microscopes in the future will reduce the amount of C16:0(d31) required to be introduced, thereby making it possible to measure the gauche/trans ratio of subcellular lipid environments under a wider range of conditions.

A Raman microscopy-based approach different from ours was recently reported to monitor the physical properties of lipids as a lateral diffusion coefficient[12]. The diffusion coefficient is commonly used to represent the physical properties of lipids, and is known to be directly related to membrane viscosity. They estimated this value by observing the ripple-like patterns formed by pulse-labeling with C16:0(d31) for one hour. In this sense, these newly discovered domains are appropriate targets for monitoring the lipid environment, as they are relatively large (around 10 μm) structures that can grow stably for several hours while continuously taking up C16:0(d31). In other words, it is unclear whether this method can be applied to other much smaller and more highly dynamic organelles. They also performed a ratiometric analysis utilizing the transition of C16:0(d31) spectra to confirm that the domains were actually phase-separated. Although this method can partially quantify the lipid environment, calculating the ratio of just two spectral data points is unstable, especially where the intensity of the C16:0(d31) signal is weak and the effect of the background signal is not negligible. In addition, it was ambiguous what aspects of the lipid environment the spectral transition represented. In contrast, our approach compensates for these issues, revealing the spectral transition is reflecting the gauche/trans ratio of incorporated C16:0(d31) at the same time.

Other approaches using Raman microscopy have been reported for characterizing the lipid composition of LDs[21–23]. In these studies, they evaluated the intensities of C–H stretching vibration of the C=C bond (i.e., =C–H stretch), C=C stretching vibration, and/or the C–H deformation vibrations to estimate the degree of unsaturation. Although lipid composition or degree of unsaturation does affect the physical properties of lipids, they do not provide direct information on the physical properties of lipids. Since membrane proteins are sometimes regulated by the physical properties of lipids rather than lipid composition, it is difficult to discuss how lipids affect the biological functions of cells based on lipid composition alone. In addition, these analyses use spectra in the fingerprint region and C–H stretch region, where many spectra of other cellular compounds as well as lipids can be observed. Therefore, it was difficult to analyze the composition or the physical properties of lipids in non-LD regions, where the

amount of lipids against other subcellular molecules should be relatively lower than LD regions. In contrast, our approach can directly measure the physical properties of lipids as the gauche/trans ratio, using the C–D vibration that appears in the silent region. Therefore, a detailed analysis of the physical properties of lipids was achieved even in the non-LD regions.

Accumulation of C16:0(d31) in LDs enabled us to investigate and identify new biological insights of LD function to buffer the physical properties of subcellular lipids. Recent studies revealed a variety of LD functions other than their classical role as an energy storage organelle. For example, it is becoming clear that LDs are involved in the initiation of signal transductions, ER stress regulator, and antioxidant roles[19,24–26]. However, as far as we know, the extensive study as a buffering role of subcellular gauche/trans ratio and the physical properties of lipids have not been documented. This is in part due to the lack of appropriate probes to study the physical properties of LDs. In our study, the gauche/trans ratio of LDs changed dramatically in response to fatty acid treatment or pharmacological intervention, while that of non-LD regions remained relatively constant (Fig. 5f, g). Meanwhile, such robustness of non-LD regions was lost when LD formation was not observed by the treatment with a high concentration of C16:0 (130 μM in total) (Fig. 5e–h). These results imply the existence of some mechanisms through which the lipid environment in the non-LD region is maintained, possibly with LDs acting as a buffer for "excessive gauche/trans ratio". Although localization patterns of incorporated labeled fatty acids in non-LD regions are unclear, previous reports including ours suggest they mainly localize to ER, where LDs are generated[12,14]. Considering that decreased unsaturation of phospholipids has been reported to antagonize the formation of LDs and lipoproteins[5,27], the loss of the robustness in the non-LD region could be due to falling into a negative spiral of decreased unsaturation in ER, initiated by the treatment with high doses of C16:0 beyond the capacity of the buffering function of LDs to maintain the gauche/trans ratio of ER and its function to generate LDs. Biological phenomena we discovered offer new insight into the functions of LDs to buffer the subcellular physical properties of lipids, which were previously mainly regarded as only an energy storage organelle.

Biochemical analyses also support this concept. Measurement using GCMS revealed that more than half of C16:0(d31) incorporated into cells was distributed to the phospholipid fraction in all conditions (Figure S6). That means cells do not just store the excess amounts of lipids only in LDs, but they also incorporate them in non-LD regions. Taken together with the fact that the total amount of C16:0 did not change significantly in the phospholipid fraction, the presence of a rapid metabolic turnover of the fatty acid moiety of phospholipids is suggested under the conditions of fatty acid treatment. These results indicate that cells may selectively degrade unnecessary lipids in non-LD regions and/or export them to LDs, so that the gauche/trans ratio of ER can be maintained. Further analysis is required to elucidate the mechanisms behind this phenomenon.

ER is an important organelle involved in the synthesis, folding, and transport of the majority of proteins. Abnormal increases in saturated fatty acids or decreases in unsaturated fatty acids are known to trigger ER stress via an unfolded protein response (UPR), subsequently leading to cell apoptosis[28]. On the other hand, increased unsaturated fatty acids in the phospholipid membrane also induce cytotoxicity in a similar manner via UPR-mediated apoptosis[29]. These reports demonstrated that an imbalance of saturated and unsaturated fatty acids leads to ER stress, and the physical property of ER membranes should be vital to maintain normal ER functions. LDs may serve as a buffer for this, and our results that gauche/trans ratio in non-LD region is constantly preserved support this concept. Since ER is involved in a wide range of cellular functions, our approach could be useful for revealing the more general mechanisms of cellular robustness to external stimuli through the physical properties of ER membranes.

There are some limitations to our methods: the assumption that the gauche/trans ratio is a dominant modulator of the spectral transitions of C–D stretch, and insufficient dynamic ranges.

For the first point, spectra derived from C–D stretch were converted into the PC18:1(d0)/PC16:0(d62) ratio, and further into the gauche/trans ratio during the quantification processes. In the process of converting the PC18:1(d0)/PC16:0(d62) ratio into the gauche/trans ratio, we used DFT to demonstrate that the spectral transitions were generated by changes in the gauche/trans ratio, and that the gauche/trans ratio was actually changed by the PC18:1(d0)/PC16:0(d62) ratio of model membranes in the MD simulation. However, it cannot be ruled out that similar spectral transitions may be induced by other factors of molecular states. For example, intermolecular interactions may also affect the vibrational state of the C–D stretch. In fact, in the spectra of water molecules, the strength of the hydrogen bonding is known to affect the shape of the Raman spectrum derived from the O–H stretch. Since the density of molecules dissolved in water affects the strength of hydrogen bonding, this phenomenon is used to measure subcellular water concentration[30]. Further studies, both in vitro and in silico, are required to estimate the physical properties of PC18:1(d0)/PC16:0(d62) phospholipid mixtures more accurately.

The second limitation to our method is the insufficient dynamic range of quantification. When comparing the fitting of spectra from LD and non-LD regions of cells treated with a total of 60 μM of C16:0 (the top two graphs in Fig. 5a), spectra fittings from LDs were slightly worse than those from the non-LD regions. The peak top of measured spectra from the LD region (black line in the top-left graph of Fig. 5a) was located more to the left than the best-fitted spectra (dashed blue line in the top-left graph of Fig. 5a), suggesting that the gauche/trans ratio of LDs in this condition was lower than that of phospholipid mixture with a PC18:1(d0)/PC16:0(d62) ratio equal to 0. This may be due to the fact that LDs predominantly consist of triglycerides and cholesterol esters, allowing them to be packed at a higher density than lipid films composed of only phospholipids. In fact, introducing cholesterols is known to reduce the fluidity of phospholipid membranes, by helping phospholipid molecules being packed with higher density. We used a pure phospholipid mixture model as the in vitro reference sample this time, but introducing triglycerides or cholesterols to the model would enable the measurement of the physical properties of lipids with higher dynamic ranges.

## Methods

**Labeled and non-labeled lipids.** C16:0(d2) (48966-2) and C18:0(d3) (49039-3) were purchased from Taiyo Nippon Sanso (Tokyo, Japan). C18:0(d5) (D-5400) was purchased from C/D/N Isotopes (Tokyo, Japan). C20:0(d4) (DLM-10519-PK) was purchased from Cambridge Isotope Laboratories (MA, USA). C16:0(d31) (16497), C18:1(d17) (9000432), C18:2(d4) (390150), C18:2(d11) (9002193), C18:3(d14) (9000433), C20:4(d8) (390010), C20:4(d11) (10006758), C20:5(d5) (10005056), C22:6(d5) (10005057), C16:0(d0) (10006627), C18:1(d0) (90260), and C20:4(d0) (90010) were purchased from Cayman Chemical (MI, USA). PC16:0(d62) (860355), PC18:1(d0) (850375), and LPC16:0(d31) (860397) were purchased from Avanti Polar Lipids (AL, USA). For C16:0(Br) and C18:0(Br), chemically synthesized compounds in a previous study were used[15].

**Cell culture.** HeLa cells were cultured in Dulbecco's modified Eagle's medium (08459; Nacalai Tesque, Kyoto, Japan) supplemented with 10% fetal bovine serum (12676-029; Thermo Fisher Scientific, Waltham, MA, USA) in a 5% CO_2 incubator. To conduct imaging experiments using labeled free fatty acids, HeLa cells were seeded on 35 mm quartz bottom dishes (SF-S-D12; Techno Alpha, Tokyo, Japan).

After 24 h of incubation, the media were replaced with the desired combination and concentration of labeled and/or non-labeled free fatty acid solutions and further incubated for 24 h. Cells were washed with phosphate-buffered saline (PBS), fixed with 4% paraformaldehyde (160-16061; Wako, Osaka, Japan) for 20 min at room temperature, and then again washed five times with PBS. To conduct the SCD1 inhibition experiment, HeLa cells were treated with an SCD1 inhibitor (ab142089; Abcam, Cambridge, United Kingdom) with the indicated concentration of free fatty acid(s) for 24 h. For disruption assay, the supernatant of fixed cells was replaced with PBS containing 50 μg/ml of digitonin (D141; Sigma Aldrich, St. Louis, MO, USA). The Raman image was acquired beforehand as the pre-treatment condition, and then another Raman image was immediately acquired after the replacement of the supernatant with digitonin for the same cells.

**Raman imaging**. To obtain Raman hyperspectral images of cells, samples were set onto a confocal Raman microscope (inVia Reflex; Renishaw, Wotton-under-Edge, United Kingdom) and excited using a 532 nm diode laser through a 63 × water immersion objective (506148, NA = 0.90; Leica, Wetzlar, Germany), followed by separation of scattered light using either 600 or 1800 line/mm grating. Exposure time and stage movement intervals were set to 0.1 s/pixel and 0.3 μm/pixel, respectively, using WiRE 5.1 software (Renishaw).

**Raman spectroscopy**. To obtain the Raman spectra of compounds, free fatty acids, LPC, and PCs were dissolved in ethanol, a mixed solution of water/ethanol (1:1 v/v), and chloroform, respectively. Aliquots (0.5 μL) of each solution, or a mixture of them, were dropped onto CaF$_2$ glass substrates (OPCFU-25C02-P; Sigma Koki, Tokyo, Japan). Following the evaporation of the solvent, Raman spectra were obtained using WiRE 5.1 software and confocal Raman spectroscopy at room temperature. Samples were excited using a 532 nm diode laser through a 50 × air objective (566072, NA = 0.75; Leica) followed by separation of scattered light using either 600 or 1800 line/mm grating.

**In silico simulation of Raman spectra**. Design and geometry optimization were performed using Avogadro version 1.2.0[31], based on molecular mechanics with a universal force field. Next, quantum chemical calculations were performed to infer Raman spectra using DFT at the B3LYP level with a 6-31pG(d) basis set in Orca version 4.2.1[32,33]. For visualization, a linear scaling factor of 0.945 and a Gaussian width of 3 were applied.

**Calculation of the reference spectra by the interpolation of Raman spectra of the PC18:1(d0)/PC16:0(d62) mixture**. The intensity of the reference spectra at the $s$-th data point along the wavenumber axis was obtained as a function of the PC18:1(d0)/PC16:0(d62) $x$, which is represented as $f_s(x)$ in the following explanation, by PCHIP-based regression. Phospholipid mixtures with the PC18:1(d0)/PC16:0(d62) ratio varying from 0 to 8 in increments of 0.5 were prepared, and their Raman spectra were measured. For each condition, 9 or 10 spectra from different positions of the samples were collected. Spectral regions ranging from 2000 to 2300 cm$^{-1}$ (210 data points with approximately 1.43 cm$^{-1}$ interval) were normalized after subtracting slope and then used for the following analysis. Then, the whole data set $U$ can be represented as follows:

$$U = \left\{ y_{x',s,N} \middle| \begin{array}{l} x' = 0.0, 0.5, \cdots, 8.0 \\ s = 1, 2, \cdots, 210 \\ N = 0, 1, \cdots, 9(, 10) \end{array} \right\}, \quad (1)$$

where $y_{x',s,N}$ represents the intensity of the normalized spectrum of the phospholipid mixture at the $s$-th data point along the wavenumber axis when the ratio of PC18:1(d0)/PC16:0(d31) and the sample number equal $x'$ and $N$, respectively. Next, PCHIP-based regression (the details will be provided in Eqs. (3–7)) was performed against sub-data set $U_{s'}$ to determine a piecewise cubic function $f_{s'}(x)$ for each $s'$ using the PchipInterpolator class of SciPy 1.2.1[34] and Python (version 3.6, https://www.python.org), where

$$U_{s'} = \left\{ y_{x',s,N} \middle| \begin{array}{l} x' = 0.0, 0.5, \cdots, 8.0 \\ s = s' \\ N = 0, 1, \cdots, 9(, 10) \end{array} \right\}. \quad (2)$$

In the following, we describe how the PCHIP-based regression was performed to determine $f_{s'}(x)$. To determine $f_{s'}(x)$, first, it was given the following two conditions. One is to pass through five coordinates $(\tilde{x}_i, \tilde{y}_{i,s'})$ called "knots" for $i = 0, 1, 2, 3, 4$, where

$$[\tilde{x}_0 \ \tilde{x}_1 \ \tilde{x}_2 \ \tilde{x}_3 \ \tilde{x}_4] = [0.0 \ 2.0 \ 4.0 \ 6.0 \ 8.0], \quad (3)$$

and $\tilde{y}_{i,s'}$ are five parameters to be optimized as discussed later in this section. The other condition is the differential coefficient of $f_{s'}(x)$ at each knot, which is represented as $\tilde{z}_{i,s'}$ and is defined using temporal variables $\tilde{w}_{j,s'} = \tilde{y}_{j,s'} - \tilde{y}_{j-1,s'}$ for $j$

= 1, 2, 3, 4 as follows:

$$\tilde{z}_{i,s'} = \begin{cases} 0 & \text{if } i = 1, 2, 3 \text{ and } \tilde{w}_{i,s'} \tilde{w}_{i+1,s'} \leq 0 \\ \dfrac{2\tilde{w}_{i,s'} \tilde{w}_{i+1,s'}}{\tilde{w}_{i,s'} + \tilde{w}_{i+1,s'}} & \text{if } i = 1, 2, 3 \text{ and } \tilde{w}_{i,s'} \tilde{w}_{i+1,s'} > 0 \\ \dfrac{3\tilde{w}_{1,s'} - \tilde{w}_{2,s'}}{2} & \text{if } i = 0 \\ \dfrac{3\tilde{w}_{4,s'} - \tilde{w}_{3,s'}}{2} & \text{if } i = 4 \end{cases}. \quad (4)$$

With these restrictions, $f_{s'}(x)$ can be represented as a piecewise cubic function as follows:

$$f_{s'}(x) = a_{j,s'} x^3 + b_{j,s'} x^2 + c_{j,s'} x + d_{j,s'} \ (\tilde{x}_{j-1} \leq x < \tilde{x}_j), \quad (5)$$

where

$$\begin{bmatrix} a_{j,s'} \\ b_{j,s'} \\ c_{j,s'} \\ d_{j,s'} \end{bmatrix} = \begin{bmatrix} (\tilde{x}_{j-1})^3 & (\tilde{x}_{j-1})^2 & \tilde{x}_{j-1} & 1 \\ (\tilde{x}_j)^3 & (\tilde{x}_j)^2 & \tilde{x}_j & 1 \\ 3(\tilde{x}_{j-1})^2 & 2\tilde{x}_{j-1} & 1 & 0 \\ 3(\tilde{x}_j)^2 & 2\tilde{x}_j & 1 & 0 \end{bmatrix}^{-1} \begin{bmatrix} \tilde{y}_{j-1,s'} \\ \tilde{y}_{j,s'} \\ \tilde{z}_{j-1,s'} \\ \tilde{z}_{j,s'} \end{bmatrix}. \quad (6)$$

Then, parameters $\tilde{y}_{i,s'}$ were optimized against $y_{x',s',N} \in U_{s'}$ to minimize the following residual sum of squares (RSS$_1$):

$$\text{RSS}_1 = \sum_{x'} \sum_N (f_{s'}(x') - y_{x',s',N})^2. \quad (7)$$

Finally, by using $f_{s'}(x)$ with the optimized coefficients $\tilde{y}_{i,s'}$, vector $\boldsymbol{y}_x$, which represents the reference Raman spectra of the phospholipid mixture with the PC18:1(d0)/PC16:0(d31) ratio equaling $x$, were generated in the range from 2000 to 2300 cm$^{-1}$ as follows:

$$\boldsymbol{y}_x = \begin{bmatrix} f_1(x) \\ \vdots \\ f_{210}(x) \end{bmatrix}. \quad (8)$$

**Conversion of Raman spectra into the PC18:1(d0)/PC16:0(d62) ratio**. The region between 2000 and 2300 cm$^{-1}$ of the measured Raman spectra at each pixel (represented by vector $\boldsymbol{m}$) was converted into virtual PC18:1(d0)/PC16:0(d62) ratio $x$ by optimizing five parameters, including $x$, ($x$, $\alpha$, $\beta$, $\gamma$, and $\delta$), to minimize the following RSS$_2$ with constraints of $\alpha$, $\beta \geq 0$:

$$\text{RSS}_2 = \|\boldsymbol{m} - (\alpha \boldsymbol{y}_x + \beta \boldsymbol{b} + \gamma \boldsymbol{s}_{(210)} + \delta \boldsymbol{i}_{(210)})\|_2^2, \quad (9)$$

Here, $\|\boldsymbol{a}\|_2$ represents the L2 norm of vector $\boldsymbol{a}$, and $\boldsymbol{b}$ represents the background spectrum in the region between 2000 and 2300 cm$^{-1}$. $\boldsymbol{s}_{(210)}$ and $\boldsymbol{i}_{(210)}$, corresponding to the slope and the intercept, respectively, are vectors with the same size as $\boldsymbol{y}_x$ (which equals 210), and are represented as follows:

$$\boldsymbol{s}_{(210)} = \begin{bmatrix} 1 \\ \vdots \\ 210 \end{bmatrix},$$

$$\boldsymbol{i}_{(210)} = \begin{bmatrix} 1 \\ \vdots \\ 1 \end{bmatrix}. \quad (10)$$

Calculations were performed using the minimize function of SciPy 1.2.1 and Python 3.6 implemented in ImageCUBE version 0.6.4[14,35].

**Background subtraction of in vivo spectra**. To simplify the comparison of the spectra, background spectra were subtracted with weighting from representative spectra of LD and non-LD regions. The weighting values $\beta_{\text{LD}}$, $\beta_{\text{nonLD}}$, $\gamma_{\text{LD}}$, $\gamma_{\text{nonLD}}$, $\delta_{\text{LD}}$, and $\delta_{\text{nonLD}}$ were calculated to minimize the following RSS values:

$$\text{RSS}_3 = \|\boldsymbol{m}'_{\text{LD}} - (\beta_{\text{LD}} \cdot \boldsymbol{m}'_{\text{BG}} + \gamma_{\text{LD}} \cdot \boldsymbol{s}_{(280)} + \delta_{\text{LD}} \cdot \boldsymbol{i}_{(280)})\|_2^2,$$
$$\text{RSS}_4 = \|\boldsymbol{m}'_{\text{nonLD}} - (\beta_{\text{nonLD}} \cdot \boldsymbol{m}'_{\text{BG}} + \gamma_{\text{nonLD}} \cdot \boldsymbol{s}_{(280)} + \delta_{\text{nonLD}} \cdot \boldsymbol{i}_{(280)})\|_2^2. \quad (11)$$

Here, $\boldsymbol{m}'_{\text{LD}}$, $\boldsymbol{m}'_{\text{nonLD}}$, and $\boldsymbol{m}'_{\text{BG}}$ represent the concatenated two sub-vectors of the spectra measured in representative LD, non-LD, and background pixels, respectively. The size of each sub-vector was 140, and the region from which sub-vectors were extracted was 1800–2000 and 2300–2500 cm$^{-1}$. Of note, the procedure for this background subtraction was used only for the visualization of spectra in Fig. 3, and not for other analyses.

**MD simulation**. Model membrane systems were created using CHARMM-GUI[36,37] with the following PC18:1(d0)/PC16:0(d62) ratio (numbers in brackets followed by the ratio represent the numbers of each phospholipid molecule): 0.0 [0/200], 0.125 [24/192], 0.25 [40/160], 0.5 [68/136], 1.0 [100/100], 2.0 [136/68], 3.0 [150/50], 4.0 [160/40], 5.0 [170/34], 6.0 [180/30], 7.0 [182/26], and 8.0 [192/24]. A

height of 22.5 nm was applied to the top and bottom of the membranes and filled with a TIP3P water model. Next, simulations were performed using the GRO-MACS 2019.3 software package[38], with CHARMM36m all-atom force field[39]. First, system energy was minimized using the steepest descent method, followed by equilibration and production steps. In the equilibration steps, the following conditions were first applied for 250 ps: NVT ensemble with an integration time step of 1 fs, and a constant temperature of 298.15 K using the Berendsen thermostat[40]. In this step, position restraints of 1000 and 400 kJ mol$^{-1}$ nm$^{-2}$ were applied every 125 ps to the phosphorus atoms of each lipid. Additionally, dihedral restraints of 1000 and 400 kJ mol$^{-1}$ rad$^{-2}$ were applied to dihedral angles, defined by the following two types of four atoms (position of atoms are indicated in brackets after the atomic symbol): C[sn-1], C[sn-3], C[sn-2], O[sn-2] (for both PC16:0[d62], and PC18:1[d0]), and C[8], C[9], C[10], and C[11] of each acyl chain (for PC18:1[d0]). Next, the system was further equilibrated in NPT ensemble equilibration for 875 ps using the Berendsen barostat with a constant pressure of 1 bar[40]. In this step, positional restraints of 400, 200, 40, and 0 (no restraints) kJ mol$^{-1}$ nm$^{-2}$, dihedral restraints of 200, 200, 100, and 0 kJ mol$^{-1}$ rad$^{-2}$, and a time step of 1, 2, 2, and 2 fs were applied for the first 125 ps and every subsequent 250 ps.

During production simulations, no restraints were applied to the systems, and the temperature and pressure were kept constant at 298.15 K and 1 bar, respectively, using Nosé–Hoover thermostat[41,42] and a Parrinello-Rahman barostat[43] with semi-isotropic pressure coupling to characteristic times of 1 and 5 ps, respectively. Throughout the simulation, all non-bonded interactions were cutoff at 1.2 nm.

**Trajectory analysis**. We used GROMACS 2019.3, Python 3.6, gnuplot (version 5.2, http://www.gnuplot.info), and PyMol (version 2.3.4, https://pymol.org/2/) for the analysis and visualization of trajectories.

**Modeling of the gauche/trans ratio of the PC18:1(d0)/PC16:0(d62) mixture**. Interconversion between gauche and trans conformers was represented schematically as follows:

$$nA + A_t \underset{k_2}{\overset{k_1}{\rightleftarrows}} nA + A_g, \tag{12}$$

$$mB + A_t \underset{k_4}{\overset{k_3}{\rightleftarrows}} mB + A_g, \tag{13}$$

where $A$, $B$, $A_g$, $A_t$, $n$, $m$, and $k_i$ ($i = 1, 2, 3, 4$) denote the PC16:0(d62), PC18:1(d0), dihedral carbon atoms in PC16:0(d62) with gauche conformation, dihedral carbon atoms in PC16:0(d62) with trans conformation, order of reaction of Eq. (12), order of reaction of Eq. (13), and reaction rate constants, respectively. Based on these reaction formulae, the following equation can be established under equilibrium:

$$k_1[A]^n[A_t] + k_3[B]^m[A_t] = k_2[A]^n[A_g] + k_4[B]^m[A_g]. \tag{14}$$

This equation can be transformed as follows to calculate the gauche/trans ratio:

$$\frac{[A_g]}{[A_t]} = \frac{k_1 + k_3 x'}{k_2 + k_4 x'}, \tag{15}$$

where

$$x' = \frac{[B]^m}{[A]^n}. \tag{16}$$

Assuming that the order of reaction was the same for Eq. (12) and Eq. (13) ($n = m$), the following equation can be obtained;

$$\frac{[A_g]}{[A_t]} = p + \frac{rx^n}{q + x^n}, \tag{17}$$

where

$$\begin{aligned} x &= \frac{[B]}{[A]}, \\ p &= \frac{k_1}{k_2}, \\ &= \frac{k_2}{k_4}, \\ r &= \frac{k_3}{k_4} - \frac{k_1}{k_2}. \end{aligned} \tag{18}$$

Thus, gauche/trans ratios were expressed with the PC18:1(d0)/PC16:0(d62) ratio $x$. Parameters $p$, $q$, and $r$ were optimized to fit the data derived from MD simulations using the minimize function of SciPy 1.2.1 and Python 3.6.

**Quantification of cell images**. All images were analyzed using ImageCUBE version 0.6.4 and Fiji/ImageJ[44,45]. At the first quantification step, images were masked based on the signal-to-noise ratio (SNR$_2$) and horizontal line artifacts (HLAs)

deriving from microscopic scans. For each pixel, SNR$_2$ was calculated as follows:

$$\mathrm{SNR}_2 = \frac{\|\alpha \boldsymbol{y}_x\|_2^2}{\mathrm{RSS}_2}. \tag{19}$$

Here, $\alpha$, $\boldsymbol{y}_x$, and RSS$_2$ are derived from Eq. (8) and Eq. (9). Images constructed using the value $\|\alpha \boldsymbol{y}_x\|_2^2$, RSS$_2$, and SNR$_2$ are shown as $\boldsymbol{I}_{\|\alpha \boldsymbol{y}_x\|_2^2}$, $\boldsymbol{I}_{\mathrm{RSS}_2}$, and $\boldsymbol{I}_{\mathrm{SNR}_2}$, respectively, in Fig. S8.

HLAs were defined for each image $\boldsymbol{I}_{[\alpha \boldsymbol{y}_x]}$ constructed based on the sum of the elements of $\alpha \boldsymbol{y}_x$ represented as $[\alpha \boldsymbol{y}_x]$ as follows:

$$\mathrm{HLAs} = \{\mathrm{I}_{x,y} | (\text{cosine similarity}) > 0.7\},$$

$$(\text{cosine similarity}) = \frac{\langle \boldsymbol{I}_{[\alpha \boldsymbol{y}_x](x-15:x+15,y-1:y+1)}, \boldsymbol{I}'_{(31,3)} \rangle}{\|\boldsymbol{I}_{[\alpha \boldsymbol{y}_x](x-15:x+15,y-1:y+1)}\|_2 \cdot \|\boldsymbol{I}'_{(31,3)}\|_2}, \tag{20}$$

where $\mathrm{I}_{x,y}$ represents the pixel at $(x, y)$, and the operations $\langle \boldsymbol{A}, \boldsymbol{B} \rangle$ and $\|\boldsymbol{A}\|_2$ represent the inner product of matrices $\boldsymbol{A}$ and $\boldsymbol{B}$, and the L2 norm of matrix $\boldsymbol{A}$, respectively. $\boldsymbol{I}_{[\alpha \boldsymbol{y}_x](x-15:x+15,y-1:y+1)}$ represents the matrix of sub-image $\boldsymbol{I}_{[\alpha \boldsymbol{y}_x]}$, with x and y coordinates ranging from x−15 to x+15 and from y−1 to y−1, respectively. $\boldsymbol{I}'_{(31,3)}$ is a matrix of the same size as $\boldsymbol{I}_{[\alpha \boldsymbol{y}_x](x-15:x+15,y-1:y+1)}$ and is described as follows:

$$\boldsymbol{I}_{(31,3)} = \begin{bmatrix} 0 & 0 & \cdots & 0 \\ 1 & 1 & \cdots & 1 \\ 0 & 0 & \cdots & 0 \end{bmatrix}. \tag{21}$$

Next, HLAs were horizontally extended for 10 pixels to acquire HLAs$_{\mathrm{extended}}$. Intermediate images of the process are shown in Fig. S8 as $\boldsymbol{I}_{\cos}$, $\boldsymbol{I}_{\mathrm{HLAs}}$, and $\boldsymbol{I}_{\mathrm{HLAsEx}}$. Finally, pixels that satisfy SNR$_2 > 100$, and not being included in HLAs$_{\mathrm{extended}}$, were subsequently analyzed ($\boldsymbol{I}_{\mathrm{MASK}}$ in Fig. S8).

As a second step, LD and non-LD areas were defined from image $\boldsymbol{I}_{[\alpha \boldsymbol{y}_x]}$. First, the background image $\boldsymbol{I}_{\mathrm{BG}}$ was generated using a median filter with a kernel size of 21, which was subsequently subtracted from image $\boldsymbol{I}_{[\alpha \boldsymbol{y}_x]}$. Next, image convolution was performed ($\boldsymbol{I}_{\mathrm{CONV}}$ in Fig. S8) to detect pixels in which the brightness of the image drastically changed, using the following kernel $\boldsymbol{K}$ with size 5:

$$\boldsymbol{K} = \begin{bmatrix} 0 & -1 & -1 & -1 & 0 \\ -1 & -1 & -1 & -1 & -1 \\ -1 & -1 & 24 & -1 & -1 \\ -1 & -1 & -1 & -1 & -1 \\ 0 & -1 & -1 & -1 & 0 \end{bmatrix}. \tag{22}$$

After convolution, the Fourier transformation (FT) bandpass filter was applied, filtering large and small structures down to 5 and up to 2 pixels, respectively ($\boldsymbol{I}_{\mathrm{FT}}$ in Fig. S8). Next, the *threshold* was set as follows using the modal intensity of $\boldsymbol{I}_{\mathrm{FT}}$:

$$threshold = \mathrm{SNR} \cdot (\text{modal intensity}), \tag{23}$$

where SNR was arbitrarily set to 5. Finally, pixels whose intensities were greater than the *threshold* and that were included in $\boldsymbol{I}_{\mathrm{MASK}}$ were defined as LD areas ($\boldsymbol{I}_{\mathrm{LD}}$ in Fig. S8). To define non-LD areas ($\boldsymbol{I}_{\mathrm{nonLD}}$ in Fig. S8), LD areas were swelled by 2 pixels ($\boldsymbol{I}_{\mathrm{SLD}}$ in Fig. S8) and subtracted from $\boldsymbol{I}_{\mathrm{MASK}}$.

Note that, as this procedure produces unexpected results when applied to cells without LDs, such cells were analyzed manually.

**Lipid extraction and fractionation**. Confluent HeLa cells were washed with PBS and collected using 0.25% trypsin-EDTA solution (T4049; Sigma Aldrich, St. Louis, MO, USA), and then lipids were extracted using the Bligh and Dyer method[46]. Briefly, the sample (1 ml mixture of 0.25% trypsin-EDTA and PBS) was mixed with 2.5 ml of methanol containing 0.03% formic acid and 1.25 ml of chloroform. The mixture was vortexed well and another 1.25 ml of chloroform and 1.25 ml of water were added. After vortexing the sample well, they were centrifuged at 1000 × g for 10 min. The bottom organic layer was collected as the lipid extract. For the lipid fractionation, neutral lipids and phospholipids were separated by solid-phase extraction using InertSep NH$_2$ aminopropyl columns (5010-61602; GL Sciences, Tokyo, Japan). Extracted lipids were dried in a centrifugal evaporator, dissolved in chloroform, and applied to the column preconditioned with 10 ml of hexane. The flow-through was collected and combined with the fraction eluted by the mixture of chloroform/isopropanol (2:1, v/v) to obtain a neutral lipid fraction. After removal of free fatty acids with diethyl ether containing 2% acetic acid, phospholipid fraction was eluted with the mixture of methanol/28% ammonia solution (016-03146; Fujifilm, Osaka, Japan) (9:1, v/v).

**Quantification of C16:0 amount by GC-MS**. Fractionated lipids were dried in a centrifugal evaporator. After adding 0.5 mg of free fatty acid C23:0(d0) (T6543; Sigma Aldrich, St. Louis, MO, USA) to the samples as an internal standard, methylation of free fatty acid and transmethylation of esterified fatty acid were performed using the Fatty Acid Methylation Kit (06482-04; Nacalai Tesque, Kyoto, Japan). Generated fatty acid methyl esters (FAME) were purified using the Fatty Acid Methyl Ester Purification Kit (06483-94; Nacalai Tesque, Kyoto, Japan) following the manufacturer's instructions, then samples were measured using GCMS

(GCMS-QP2010 Ultra; Shimadzu, Kyoto, Japan) equipped with FAMEWAX fused silica capillary column (30 m × 0.25 mm, internal diameter × 0.25 μm) (12497; Restek, Bellefonte, PA, USA). An aliquot of 1 μL or 4 μL-sample was injected in splitless mode, with helium used as the carrier gas at a flow rate of 45 cm/s (linear velocity). The injection port temperature was set at 250 °C. The oven temperature was set to 40 °C and maintained for 2 min, raised to 140, 200, and 240 °C at a rate of 20, 11, and 3 °C/min, respectively, and finally held at 240 °C for 10 min. The interface and ion source temperatures were maintained at 250 and 200 °C, respectively. The mass spectrometer was operated in electron impact ionization mode with 70 eV ionization energy, and the mass spectra were obtained in scan mode from $m/z$ 50 to 600 with the cycle time set to 0.3 s. A 37-component FAME mix standard (CRM47885; Sigma Aldrich, St. Louis, MO, USA) was used to identify and quantify C16:0(d0) and C23:0(d0) methyl esters. For the identification and the quantification of C16:0(d31) methyl ester, the known concentrations of C16:0(d31) and C23:0(d0) were mixed, methylated, purified, measured by GCMS, and used as a standard. The area under the curve of the mass chromatograms with $m/z$ of 74, 301, and 74 were used to calculate the amounts of C16:0(d0), C16:0(d31), and C23:0(d0) in samples.

**Statistical analysis**. All statistical analyses were performed using Python 3.6 (https://www.python.org).

**Reporting summary**. Further information on research design is available in the Nature Research Reporting Summary linked to this article.

## Data availability
The data that support the findings of this study are available from the corresponding author upon request. Source data for the figures are available as supplementary data files and any remaining information can be obtained from the corresponding author upon reasonable request.

## Code availability
The Python, ImageCUBE, and ImageJ code[35,47] is available from GitHub (https://github.com/MasaakiU/ImageCUBE and https://github.com/MasaakiU/ImageJ_macro) or the corresponding author upon request.

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

## Acknowledgements

This work was supported by AMED-CREST 20gm0910011 (to Hideo Shindou, Department of Lipid Signaling, National Center for Global Health and Medicine), AMED Program for Basic and Clinical Research on Hepatitis JP20fk0210041 (to Hideo Shindou), and Japan Society for the Promotion of Science KAKENHI Grant-in-Aid for JSPS Fellows 18J21897 (to Masaaki Uematsu). Takao Shimizu is supported by the AMED GAPFREE Program (19ak0101043h0005) and the Takeda Science Foundation. Department of Lipid Signaling, National Center for Global Health and Medicine is supported by Ono Pharmaceutical Company, Japan. We would like to express our gratitude to Hideo Shindou for providing a good research environment and for a thorough discussion. We also thank Hiroshi Noguchi for providing valuable advice on MD simulation, Saori Uematsu, Keisuke Yanagida, and Daisuke Hishikawa for their thoughtful discussions and review of the manuscript, Shota Yamamoto, Tomoyuki Suzuki, Natsuko F. Inagaki, Miyuki Yamamoto, Yukiko Sugimoto, and Megumi Yasuda for kindly allowing us to use their computers for the analyses. All MD simulation results in this paper were generated on the NIG supercomputer at ROIS National Institute of Genetics, Japan.

## Author contributions

M. Uematsu conceptualized the project, developed the analytical methods, performed the in vivo, in vitro, and in silico experiments, investigated the data, and wrote the paper. T. Shimizu gave a suggestion for the conceptualization of the project, procured research equipment, edited the paper, and administered the project.

## Competing interests

The authors declare the following competing interests; M. Uematsu and T. Shimizu are members of the Department of Lipidomics, Graduate School of Medicine, The University of Tokyo, which is financially supported by the Shimadzu Corporation; M. Uematsu and T. Shimizu are members of the Department of Lipid Signaling, National Center for Global Health and Medicine, which is conducting joint research with Ono Pharmaceutical Company. The authors declare no other competing interests.
