## [Transparent Peer Review File · Communications Biology]

Reviewers' comments:

Reviewer #1 (Remarks to the Author):

The authors demonstrate a systematic framework to quantify the gauche/trans ratio in long alkyl chains of fatty acids. Such ratio is believed to be reflective of local lipid environment. By replacing the C-H bonds on fatty acids with C-D bonds, the authors compared C-D Raman spectra of fatty acids in vitro with those after being up-taken into HeLa cells. Their simulation results imply that the change in C-D spectra originate from the change in gauche/trans ratio in alkyl chains. This spectral change can be re-capitulated by in vitro mixture of saturated and unsaturated phosphatidylcholine with different ratios. The phosphatidylcholine ratio can be further converted to gauche/trans ratio of fatty acid using molecular dynamic simulation quantitatively. The authors then applied this method to understand the difference between lipid droplets and non-lipid droplets lipid in HeLa cells and found that environments of lipid droplets fluctuate more compared to that of other lipids.

Although the authors present a rather convincing and quantitative method to gauche/trans ratio of fatty acid, the biological conclusions they drew from the observation need more evidence. I would recommend reconsidering the manuscript after major revision. Here are my concerns:

1. Is it possible that the observed relative unchanged gauche/trans ratio a result of slow lipid metabolic rate in non-LD region? As HeLa cells tend not to form lipid droplets under native condition, its natural the extracellular lipids accumulate first in LD and slowly exchange with other cellular compartments. If the incubation time is only 24h, the unsaturated lipids only got enough time to accumulate in LD, not to incorporate into membranes. This also explains when saturated lipid concentration increased a lot, the authors observed a much lower gauche/trans ratio in non-LD region as well (Figure 5F, G). Similarly, the SCD1 inhibition is too short (1h). The difference in gauche/trans ratio after inhibition was barely seen.

2. Authors should have proper experimental data and discussion to support of the physical implication of gauche/trans ratio. Although they referred previous literature and said gauche/trans ratio is indicative of viscosity, the paper didn't present any experimental data, such as Raman spectra of C-D fatty acids dissolving in various ratios of water glycerol mixtures. Besides, the authors only briefly mentioned that the packing effect of fatty acids could also contribute to the spectral change. The packing effect should be discussed in more detail. I can argue that packing difference is a more direct consequence of changing gauche/trans ratio, since the melting points of pure fatty acids is directly related to the packing of the molecules.

3. Disruption assay can be helpful to strengthen the claim by the authors. Triton-wash, or hexanediol treatment can be used to find even higher gauche/trans ratio condition, providing the work a more comprehensive view of lipid environment.

Minor points:

Figure 1 B. Showing the C-Br Raman spectra is not convincing as the whole manuscript is talking about C-D spectra. The authors can mix C 16:0 Br with C 16:0 d31 to get liquid state saturate fatty acid. Then can compare the spectra of C-D rather than showing C-Br.

Figure 1 C, D. I think the C-D spectra should be shown rather than full Raman spectra.

Figure 5 and Figure 3 are basically the same experiments, with or without quantification. Showing Figure 5 is already sufficient.

Writing can be improved by presenting the data in a logical order. Start from in vitro to in silico to in vivo. Current version is all over the place.

Reviewer #2 (Remarks to the Author):

Uematsu and Shimizu have used Raman spectroscopy to quantify changes in lipid droplets (LDs) and in non-LD cell regions with change in saturated and unsaturated fatty acid availability in culture medium. From their results they suggest that Raman spectroscopy can yield a quantitative measure of changes in lipid conformation. They further suggest that they can quantify trans / gauche ratios in lipid chains. They also draw some broad biological conclusions based on their data.

In figure 5, the authors show a quantitative measure of four or five Raman spectral for each of several culture conditions having different ratios of saturated and unsaturated fatty acids in the medium. The spectra from LD regions and non-LD regions appear to have statistically significant trends with changes in culture conditions, supporting the idea that Raman spectra reflect changes in LD composition that are not apparent in the non-LD regions.

Further, DFT calculations suggest slight differences in a shoulder at 2130 wavenumbers with changes in trans / gauche ratio. The authors do indeed see changes in the spectra near 2130 wavenumbers. As suggested in the calculations, the differences are subtle, but the results appear robust.

The three primary issues I have with the work are A) the assignment of quantitative trans - gauche ratios to the Raman spectra obtained in vivo, and B) the biological conclusions regarding new insights and C) the context of the work with other Raman spectroscopy and imaging of LDs. See below for elaboration of these issues.

1) LDs have long been thought to be storage vehicles for neutral lipids (Guo, et al., Lipid droplets at a glance. *J Cell Sci* 122, 749–752 (2009).), so is it a novel hypothesis that they serve as buffers for unsaturated fats? The paper is missing important references to the biology of lipid droplets. Many reviews have now been written, and a more thorough discussion of what is known about the function of these should be included. Without some discussion of what the current biological questions or opportunities are, it is hard to judge the importance of the work.

2) Simulations using 18:1 and 16:0 mixtures were for membranes, but measurements were made on lipid droplets. LDs, which are composed of a neutral lipid core for TAGs and sterol esters with a phospholipid monolayer. The surface (monolayer) to bulk ratio is incredibly tiny (< 1:500 even for the smallest LDs they could measure (~1µm), so the Raman spectra are essentially all from the bulk. It is unclear how useful membrane simulations are in determining trans-gauche ratios in the bulk.

4) Raman spectroscopy has been used many times to characterize lipid droplets in vivo, and some of this work should be mentioned. The authors cite one study (on membranes), but they should also put their work on Raman spectroscopy of LDs in context. A representative, but by no means exhaustive list follows: (Bonn et al., Imaging of chemical and physical state of individual cellular lipid droplets using multiplex CARS microscopy. *J. Raman Spectrosc.* 40, 763–769 (2009), Chen, et al. Spectroscopic coherent Raman imaging of *Caenorhabditis elegans* reveals lipid particle diversity. *Nature Chemical Biology* 2011 7:3 3, 1–9 (2020), Schie, et al. Direct comparison of fatty acid ratios in single cellular lipid droplets as determined by comparative Raman spectroscopy and gas chromatography. *The Analyst* 138, 6662–6670 (2013).)

Responses to the comments of Reviewer #1

We are grateful for the comments and suggestions.

Specific Points:

1. Is it possible that the observed relative unchanged gauche/trans ratio a result of slow lipid metabolic rate in non-LD region? As HeLa cells tend not to form lipid droplets under native condition, its natural the extracellular lipids accumulate first in LD and slowly exchange with other cellular compartments. If the incubation time is only 24h, the unsaturated lipids only got enough time to accumulate in LD, not to incorporate into membranes. This also explains when saturated lipid concentration increased a lot, the authors observed a much lower gauche/trans ratio in non-LD region as well (Figure 5F, G). Similarly, the SCD1 inhibition is too short (1h). The difference in gauche/trans ratio after inhibition was barely seen.

<Response>

Thank you for your critical comment. As the reviewer pointed out, there is a possibility that the majority of extracellular lipids taken up into cells localize in LDs, resulting in the robust gauche/trans ratio in non-LD regions. In order to check that, we biochemically analyzed the amount of C16:0 in HeLa cells treated with the same conditions as Figs. 5F, G, and H (Fig. S7). Briefly, extracted lipids from cells were separated into the neutral lipid and phospholipid fractions, and the amount of C16:0(d0) and C16:0(d31) in each form of lipids were measured using gas chromatograph-mass spectrometry (GCMS). While LDs contain phospholipids only in a monolayer on its surface, non-LD regions contain many tubular and vesicular membrane structures composed of phospholipid bilayers. Therefore, the C16:0 amount in phospholipid fraction should mainly reflect that in non-LD regions.

As a result, we found that more than half of C16:0(d31) taken up into cells was distributed to the phospholipid fraction for every condition. In addition, the total amount of C16:0 in the phospholipid fraction was not changed significantly, while that in the neutral lipid fraction was dramatically altered for conditions treated with more than 30 μ M of C16:0(d31). These results exclude the possibility that the robustness in non-LD regions is achieved by regulating the distribution of incorporated fatty acids only in LD regions and by preventing their translocation to non-LD regions. Instead, cells may actively incorporate external fatty acids into both LD and non-LD regions, and the robustness in non-LD regions may be maintained through the regulation of lipid exchange between LD and non-LD regions. We updated the main manuscript (Fig. S7 and P.12 [l.355], P15 [l.446] in the revised manuscript).

About the SCD1 inhibition experiments, we originally planned to pretreat HeLa cells with SCD1 inhibitor for 1 h and then further incubate them in the media containing fatty acids and SCD1 inhibitor. That means the total of 25 h incubation with SCD1 inhibitor. However, it turned out that we did not perform the pre-treatment with SCD1 inhibitor (i.e., just 24 h treatment with fatty acids) for the data we used in the figures. This is a mistake in our description, and we corrected the materials and method section (P.17 [l.527] in the revised manuscript). 24 h of incubation is not too short for SCD inhibition.

2. Authors should have proper experimental data and discussion to support of the physical implication of gauche/trans ratio. Although they referred previous literature and said gauche/trans ratio is indicative of viscosity, the paper didn't present any experimental data, such as Raman spectra of C-D fatty acids dissolving in various ratios of water glycerol mixtures.

<Response>

Thank you for the constructive comment. As the reviewer pointed out, dissolving C–D fatty acids in the glycerol/water mixture is a promising experiment. However, lipids are not completely dissolved in the glycerol/water mixture. In fact, even when we mix C16:0(d31) with C18:1(d0), both of them are fatty acids, in Fig. S4A, they were not completely mixed up, and the relative amount of C16:0(d31) to C18:1(d0) was not uniform (please also refer to Figures only for reviewers 1).

Although the solvent is not glycerol/water mixture, Figs. 1C, 1D, 1E, 4B, S4A, and S4B (old Figs. 1C, 1D, 4B, S4A, and S4B) are considered to be equivalent experiments. In these figures, we mixed lipids containing C–D fatty acid (C16:0[d31], LPC16:0[d31], and PC16:0[d62]) with lipids containing C18:1(d0) fatty acid (C18:1[d0] and PC18:1[d0]). The viscosities of lipid environments are affected by the degree of unsaturation in the fatty acid moiety of the phospholipid molecules. Therefore, when these lipids are mixed in various proportions, the viscosities of the lipid environments are changed, and the results show the corresponding transitions of Raman spectra in our results.

Besides, the authors only briefly mentioned that the packing effect of fatty acids could also contribute to the spectral change. The packing effect should be discussed in more detail. I can argue that packing difference is a more direct consequence of changing gauche/trans ratio, since the melting points of pure fatty acids is directly related to the packing of the molecules.

<Response>

As the reviewer rightly pointed out, the relationship between the gauche/trans conformation and the viscosity is indirect. We agree to state more clearly about the roles of packing effect linking gauche/trans conformation and the viscosity. We updated the manuscript (P.6 [l.157] in the revised manuscript).

3. Disruption assay can be helpful to strengthen the claim by the authors. Triton-wash, or hexanediol treatment can be used to find even higher gauche/trans ratio condition, providing the work a more comprehensive view of lipid environment.

<Response>

As the reviewer and editor suggested, disruption assay is very helpful to our study, and we performed the experiments. We treated HeLa cells with weak detergent (50 $\mu\text{g}/\text{mL}$ digitonin) to loosen the lipid packing and increase the gauche/trans ratio, without destroying subcellular lipid structures. The results of the Raman imaging showed the increased PC18:1(d0)/PC16:0(d62) and gauche/trans ratios in non-LD regions by the digitonin treatment (Fig. S6). These results suggest that the observed spectral

transitions actually reflect changes in the physical properties of the lipids, in particular increases in the gauche/trans ratio caused by the weakened lipid packing. In contrast, PC18:1(d0)/PC16:0(d62) and gauche/trans ratios did not change in LD regions, suggesting that the LD regions are more resistant to external environmental changes than the non-LD regions. The results and the discussions were added to the main texts (P.10 [l.291] in the revised manuscript). Thank you for advising important experiments. This has indeed strengthened our study.

Minor points:

Figure 1 B. Showing the C-Br Raman spectra is not convincing as the whole manuscript is talking about C-D spectra. The authors can mix C 16:0 Br with C 16:0 d31 to get liquid state saturate fatty acid. Then can compare the spectra of C-D rather than showing C-Br.

<Response>

Thank you for the comment. As the reviewer pointed out, showing the C–Br Raman spectra is not convincing. Since we are focusing on the C–D spectra in our paper, we switched the Fig. 1B and Fig. S1A. Of note, when C16:0(Br) is mixed with C16:0(d31) *in vitro*, transition of the C–D spectra were observed in the similar way as Fig. 1E and Fig. S4A (Figures only for reviewers 2).

Figure 1 C, D. I think the C-D spectra should be shown rather than full Raman spectra.

<Response>

Since the intensity of the spectra is relative for each sample, we need to show the spectrum derived from C–H vibration of 18:1(d0) to roughly adjust the intensity. Therefore, the whole spectrum (2000-3100 cm^{-1}) was displayed in the figure. However, to make it easier to compare with other figures (such as Fig. 3, 4, and 5, etc.), we also added the figure showing only the region of C–D spectra (2000-2300 cm^{-1}) as Fig. 1E.

Figure 5 and Figure 3 are basically the same experiments, with or without quantification. Showing Figure 5 is already sufficient. Writing can be improved by presenting the data in a logical order. Start from *in vitro* to *in silico* to *in vivo*. Current version is all over the place.

<Response>

Thank you for the comment. Fig. 3 and Fig. 5 are basically the same, with or without quantification. However, we need to show representative *in vivo* spectra in Fig. 3 before generating the reference spectra necessary for the quantification method in Fig. 4, because these representative *in vivo* spectra have to be displayed in Fig. 4B (indicated with orange and blue lines). Although the reference spectra themselves are generated based only on the *in vitro* spectra and does not contain any information about *in vivo* spectra, the reference spectra should mimic the *in vivo* spectra. Therefore, we chose *in vitro* lipid mixture that best represents the shape of *in vivo* spectra (that is the mixture of PC18:1[d0] and PC16:0[d62]). To discuss in

this way, we had to show *in vivo* spectra in Fig. 3 before proceed to the quantification method in Fig. 4 and the results of quantification in Fig. 5.

Moving Fig. 3 to the supplementary figures could be one option, but we would prefer not to do so because we want to clearly state that C16:0(d31) spectrum is sensitive enough to detect the physical properties of different lipid environments *in vivo* before we develop a quantification method.

Responses to the comments of Reviewer #2

We are grateful for the comments and suggestions.

1) LDs have long been thought to be storage vehicles for neutral lipids (Guo, et al., Lipid droplets at a glance. J Cell Sci 122, 749–752 (2009).), so is it a novel hypothesis that they serve as buffers for unsaturated fats? The paper is missing important references to the biology of lipid droplets. Many reviews have now been written, and a more thorough discussion of what is known about the function of these should be included. Without some discussion of what the current biological questions or opportunities are, it is hard to judge the importance of the work.

<Response>

As the reviewer pointed out, recent studies showed a variety of LD functions over their classical role as an energy storage organelle. For example, LDs can participate in signaling precursors, ER stress regulator, storage of vitamins, antioxidant role, etc^{1,2}. However, as far as we know, the extensive study as a buffering role of subcellular gauche/trans ratio and the physical properties of lipids have not been documented. We, therefore, in the revised manuscript, described what has been known of the functions of LDs citing several previous literatures, and more clearly focused on our present findings to strengthen the role of LDs to buffer gauche/trans ratio (P.1 [l.15], P.4 [l.94], and P.14 [l.423] in the revised manuscript). We also included the results of our biochemical experiments to infer the mechanisms of the new function of LDs (Fig. S7, P.12 [l.355], and P.15 [l.446] in the revised manuscript).

2) Simulations using 18:1 and 16:0 mixtures were for membranes, but measurements were made on lipid droplets. LDs, which are composed of a neutral lipid core for TAGs and sterol esters with a phospholipid monolayer. The surface (monolayer) to bulk ratio is incredibly tiny (< 1:500 even for the smallest LDs they could measure (~1µm), so the Raman spectra are essentially all from the bulk. It is unclear how useful membrane simulations are in determining trans-gauche ratios in the bulk.

<Response>

As the reviewer pointed out, the Raman spectra of LDs should mainly come from their neutral lipid core, and there should be little effect of the phospholipid monolayer. However, we do not think it is a problem, since we assume that high spectral similarities indicate a high degree of resemblance in the conformational state of C16:0(d31) (which is written in P.9 [l.251] in the revised manuscript). In other words, if the conformational state of molecules is different, they should emit the different Raman spectra.

When we measured Raman spectra of C16:0(d31) incorporated into LDs (blue and orange lines in Fig. 4B), shapes of the spectra were very similar with those of *in vitro* phospholipid mixtures of PC16:0(d62)

and PC18:1(d0) (black lines in Fig. 4B). These high spectral similarities indicate a high degree of resemblance in the conformational state (that is, the gauche/trans ratio) of C16:0(d31) incorporated into LDs *in vivo* and that of fatty acyl chain of PC16:0(d62) *in vitro*. Therefore, there is a decent reason for analyzing the Raman spectra of LDs using *in vitro* and *in silico* phospholipid mixture model.

Of note, the limitation of the above discussion is the assumption that the gauche/trans ratio is a dominant modulator of the spectral transitions of C–D vibration (which is written in P.15 [l.469] in the revised manuscript). Other states of the molecule (e.g. the strength of the intermolecular forces) can be affecting the shape of the Raman spectra. If these other factors were to precisely cancel out the effect of the gauche/trans ratio on the spectral transitions, the same Raman spectra might be observed even though the gauche/trans ratio of C16:0(d31) is different. This is one of the limitations of our present analysis, as described in the limitation section in the main manuscript (P.15 [l.469] in the revised manuscript).

In the future, this study can be improved by conducting *in vitro* and *in silico* analyses of neutral lipids in the similar way as those of phospholipids that we did in this study. However, such analyses are currently challenging, because we do not know how TAGs assemble in LDs and therefore it is difficult to perform *in silico* simulation of neutral lipids to calculate the gauche/trans ratio. Previous reports suggest that the TAG with three C16:0(d0) are in the form of the letter “h” or “tuning fork”³, but there is no consensus yet. The development of both *in vivo* and *in silico* understandings on TAGs are desirable in the future.

4) Raman spectroscopy has been used many times to characterize lipid droplets *in vivo*, and some of this work should be mentioned. The authors cite one study (on membranes), but they should also put their work on Raman spectroscopy of LDs in context. A representative, but by no means exhaustive list follows: (Bonn et al., Imaging of chemical and physical state of individual cellular lipid droplets using multiplex CARS microscopy. *J. Raman Spectrosc.* **40, 763–769 (2009), Chen, et al. Spectroscopic coherent Raman imaging of *Caenorhabditis elegans* reveals lipid particle diversity. *Nature Chemical Biology* **2011** 7:3 3, 1–9 (2020), Schie, et al. Direct comparison of fatty acid ratios in single cellular lipid droplets as determined by comparative Raman spectroscopy and gas chromatography. *The Analyst* **138**, 6662–6670 (2013).)**

<Response>

Thank you for the comment and reference suggestions. As the reviewer pointed out, there are many studies that characterized LDs using Raman microscopy. We cited previous studies that focused on the characteristics of LDs using Raman microscopy (P.13 [l.407] in the revised manuscript).

Summary of major changes made in the revised manuscript

- Added description about the packing effect (P.6 [l.157] in the revised manuscript).
- Added the results of disruption assay (Fig. S6 and P.10 [l.291] in the revised manuscript).

- Added the results of biochemical analysis (Fig. S7, P.12 [l.355], and P.15 [l.446] in the revised manuscript).
- Added the discussion about previous studies on LDs using Raman spectroscopy (P.13 [l.407] in the revised manuscript).
- Added the discussion about the previous studies on the functions of LDs (P.14 [l.423]).
- Corrected the description about the SCD1 inhibitor treatment (P.17 [l.527] in the revised manuscript).
- Added the materials and methods for lipid extraction and fractionation (P.24 [l.714] in the revised manuscript).
- Added the materials and methods for quantification of C16:0 amount by GCMS (P.24 [l.731] in the revised manuscript).
- Figures were added or re-ordered as follows:

New figure numbers	Old figure numbers	Note
Fig. 1A, C, D	Fig. 1A, C, D	no change
Fig. 1B	Fig. S1A	re-ordered
Fig. 1E	–	newly added
Fig. 2, 3, 4, 5	Fig. 2, 3, 4, 5	no change
Fig. S1A	Fig. 1B	re-ordered
Fig. S1B	Fig. S1B	no change
Fig. S2, S3, S4, S5	Fig. S2, S3, S4, S5	no change
Fig. S6, S7	–	newly added
Fig. S8	Fig. S6	re-ordered

Other minor corrections that are not displayed with red letters in the main manuscript:

- Added the line numbering in the main manuscript.
- Changed the legend style of the crosshairs in Figs. 1A, 1B, 1C, 3A, 3B, and S1A
- Changed the color of the bar graph in Fig. 5H
- Changed the colormap of the images in Figs. 5A, B, C, D, and E
- Added the “Statistical analysis” section in the materials and methods (P.25 [l.756] in the revised manuscript).
- Changed the description about the code availability in the materials and methods (P.25 [l.759] in the revised manuscript).

References for the rebuttal letter

1. Fujimoto, T. & Parton, R. G. Not just fat: The structure and function of the lipid droplet. *Cold Spring Harb. Perspect. Biol.* **3**, 1–17 (2011).
2. Welte, M. A. & Gould, A. P. Lipid droplet functions beyond energy storage. *Biochim. Biophys. Acta - Mol. Cell Biol. Lipids* **1862**, 1260–1272 (2017).
3. Pink, D. A. *et al.* Modeling the solid-liquid phase transition in saturated triglycerides. 1–11 (2010). doi:10.1063/1.3276108

Again, we want to thank the reviewers for their constructive criticism. We feel that the thoroughly revised manuscript significantly gained quality.

Sincerely yours,

Masaaki Uematsu, M.D., Ph.D.

Department of Lipidomics, Graduate School of Medicine, The University of Tokyo

7-3-1, Hongo, Bunkyo-ku, Tokyo, 113-8654, Japan

TEL: +81-3-5841-3540

Fax: +81-3-5841-3544

Lipid Signaling Project, National Center for Global Health and Medicine

1-21-1, Toyama, Shinjuku-ku, Tokyo, 162-8655, Japan

TEL: +81-3-5273-5351

Fax: +81-3-3202-7364

e-mail:

uematsu@m.u-tokyo.ac.jp

REVIEWERS' COMMENTS:

Reviewer #1 (Remarks to the Author):

The authors have addressed my concerns.

Reviewer #2 (Remarks to the Author):

The authors have addressed my concerns.